# HUMAN-ORIENTED REPRESENTATION LEARNING FOR ROBOTIC MANIPULATION

## ABSTRACT

Humans inherently possess generalizable visual representations that empower them to efficiently explore and interact with the environments in manipulation tasks. We advocate that such a representation automatically arises from simultaneously learning about multiple simple perceptual skills that are critical for everyday scenarios (e.g., hand detection, state estimate, etc.) and is better suited for learning robot manipulation policies compared to current state-of-the-art visual representations purely based on self-supervised objectives. We formalize this idea through the lens of human-oriented multi-task fine-tuning on top of pre-trained visual encoders, where each task is a perceptual skill tied to human-environment interactions. We introduce Task Fusion Decoder as a plug-and-play embedding translator that utilizes the underlying relationships among these perceptual skills to guide the representation learning towards encoding meaningful structure for what's important for all perceptual skills, ultimately empowering learning of downstream robotic manipulation tasks. Extensive experiments across a range of robotic tasks and embodiments, in both simulations and real-world environments, show that our Task Fusion Decoder improves the representation of three state-of-the-art visual encoders including R3M, MVP, and EgoVLP, for downstream manipulation policy-learning. More demos, datasets, models, and code can be found at our anonymous webpage.

## 1 INTRODUCTION

In the fields of robotics and artificial intelligence, imbuing machines with the ability to efficiently interact with their environment has long been a challenging problem. While humans can effortlessly explore and manipulate their surroundings with very high generalization, robots often fail even when faced with basic manipulation tasks, particularly in unfamiliar environments. These representations empower us to perceive and interact with our environment, effectively learning complex manipulation skills. How to learn generalizable representations for robotic manipulations thus has drawn much attention.

Existing representation learning for robotics can be generally divided into three streams. **1)** Traditionally representations were hand-crafted (*e.g.*, key point detection (Das et al., 2021) inspired by biological studies (Johansson, 1973)). They provide strong inductive bias from human engineers, but encode a limited understanding of what matters about human behavior. **2)** Modern state-of-the-art methods (Chen et al., 2016; Higgins et al., 2016; He et al., 2020; Chen et al., 2020; He et al., 2022; Nair et al., 2022) propose to automatically discover generalizable representations from data, *e.g.*, by masked image modeling and contrastive learning techniques. Though general-purpose or language semantic-based representations can be learned, they fail to grasp human behavior biases and motion cues, *e.g.*, hand-object interaction, for robotic manipulation tasks. **3)** Recent human-in-the-loop methods (Bajcsy et al., 2018; Bobu et al., 2022; 2023a) attempt to disentangle and guide aspects of the representation through additional human feedback. However, they are limited to learning from low-dimensional data (*e.g.*, physical state trajectories) due to the huge amount of human labels that are required. Each of these approaches comes with its own set of drawbacks, which lead to suboptimal performance in robotic manipulations.

In this work, we propose that a robust and generalizable visual representation can be automatically derived from the simultaneous acquisition of multiple simple perceptual skills that mirror those crit-

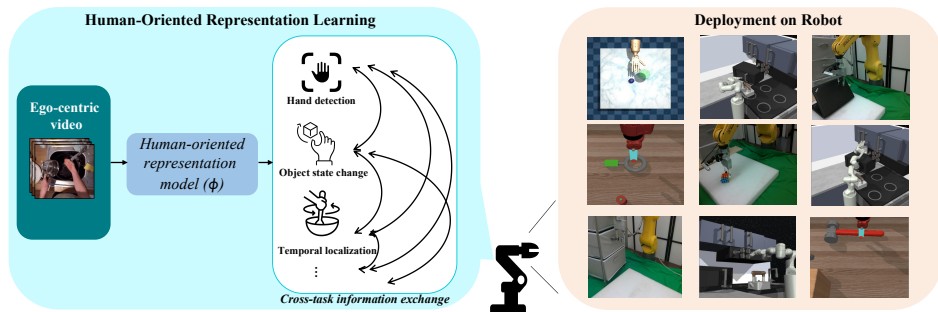

Figure 1: Left: human-oriented representation learning as a multi-task learner. Right: robots leverage the human-oriented representation to learn various manipulation tasks.

ical to human-environment interactions, as shown in Fig. 1. This concept aligns with insights from cognitive science (Kirkham et al., 2002), which posits that humans learn to extract a generalizable behavioral representation from perceptual input by mastering a multitude of simple perceptual skills, such as spatial-temporal understanding and hand-object contact estimation, all of which are critical for everyday scenarios. Centered on these human-inspired skills, we introduce Task Fusion Decoder (TFD) as a plug-and-play multitask learner to learn human-oriented representation for robotic manipulation. Unlike current state-of-the-art visual representations, which primarily rely on self-supervised objectives, our approach harnesses the power of these human-inspired perceptual skills with low-cost human priors.

Task Fusion Decoder is carefully designed with the following considerations. **1)** It learns perceptual skills on the largest ego-centric video dataset Ego4D (Grauman et al., 2022) with three representative tasks that capture how humans manipulate objects: object state change classification (OSCC), point-of-no-return temporal localization (PNR), and state change object detection (SCOD). In this way, the robot manipulation representation space is learned and distilled from real-world human experience. **2)** It takes advantage of its inside self- and cross-attention mechanisms to establish information flow across tasks through the attention matrix and learn inherent task relationships automatically through end-to-end training. The underlying relationships between these perceptual skills are utilized to guide the representation learning towards encoding meaningful structure for manipulation tasks. **3)** It is plug-and-play and can be directly built on previous foundational backbones with an efficient fine-tuning strategy, which enables it to be easily generalized and transferred to novel settings and models. We will show it improves the performance of various state-of-the-art models on various robot manipulation benchmarks and tasks.

Our contributions are three-fold. **1)** We introduce an efficient and unified framework, Task Fusion Decoder, tailored as a human-oriented multitask learner aimed at cultivating representations guided by human-inspired skills for robotic manipulations. **2)** The plug-and-play nature of our framework ensures flexibility, allowing it to seamlessly adapt to different base models and simulation environments. To demonstrate its real-world applicability, we also collect and open-source a real-world robot manipulation dataset, comprising 17 kinds of tasks featuring expert demonstrations. **3)** Extensive experiments across various model backbones (*i.e.*, MVP (Xiao et al., 2022), R3M (Nair et al., 2022), and EgoVLP (Qinghong Lin et al., 2022)), benchmarks (*i.e.*, Franka Kitchen (Gupta et al., 2019), MetaWorld (Yu et al., 2020), Adroit (Rajeswaran et al., 2017), and real-world manipulations), and diverse settings (*e.g.*, different cameras and evaluation metrics) demonstrate our effectiveness.

## 2 RELATED WORK

**Representation learning for robotic learning.** Representation learning, with the goal of acquiring effective visual encoders (Nair et al., 2022; Mu et al., 2023a; Hansen et al., 2022; Ze et al., 2023; Parisi et al., 2022; Yen-Chen et al., 2020; Shridhar et al., 2022; Khandelwal et al., 2022; Shah & Kumar, 2021; Seo et al., 2022), is crucial to computer vision and robotic learning tasks. Recently, it has been dominated by unsupervised and self-supervised methods (Chen et al., 2016; Higgins et al., 2016; He et al., 2020; Chen et al., 2020; He et al., 2022; Nair et al., 2022; Ma et al., 2022; Brohan et al., 2022; Alakuijala et al., 2023; Karamcheti et al., 2023; Mu et al., 2023b; Jing et al., 2023;

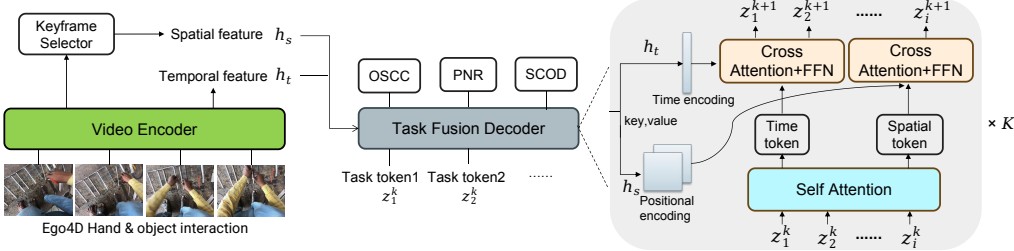

Figure 2: The pipeline for the finetuning framework by using task fusion network. The task fusion decoder which includes the cross-attention and self-attention, can adjust the video encoder representation and fuse different tasks information.

Majumdar et al., 2023). These methods try to learn disentangled representations from large datasets (Russakovsky et al., 2015; Goyal et al., 2017; Damen et al., 2018; Shan et al., 2020; Grauman et al., 2022). Though requiring little human cost, these methods purposefully bypass human input, consequently, the learned representations are prone to spurious correlations and do not necessarily capture the attributes that are important for downstream tasks (LeCun, 2022; Bobu et al., 2023b). For example, Xiao et al. (Xiao et al., 2022) propose using masked autoencoders (MAE) to learn a mid-level representation for robot learning of human motor skills (e.g., pick and place). However, the MAE representation is tailored for reconstructing pixel-level image structure and does not necessarily encode essential high-level behavior cues such as hand-object interaction. To mitigate this, another line of works attempts to leverage human priors by explicitly involving a human in the learning loop to iteratively guide the representation towards human-orientated representations (Bobu et al., 2021; Katz et al., 2021; Bobu et al., 2022; 2023a). However, these methods do not scale when learning from raw pixels due to the laborious human costs. Our idea fills the gap between unsupervised/self-supervised and human-guided representation learning. Our human-oriented representation arises from simultaneously learning about multiple perceptual skills from large and well-labeled video datasets that already capture human priors. Through this, we can effectively capture important attributes that are important for human motor skills in everyday scenarios in a human-oriented but label-efficient way.

**Multitask learning.** Multitask representation learning uses proxy tasks to instill human's intuition on important attributes about the downstream task in representation learning (Brown et al., 2020; Yamada et al., 2022). The hope is that by learning a shared representation optimized for all the tasks, robots can effectively leverage these representations for novel but related tasks. Tasks have inherent relationships and encoding their relationships into the learning process can promote generalizable representations that achieve efficient learning and task transfer (e.g., Taskonomy (Zamir et al., 2018) and Cross-Task (Zamir et al., 2020)). However, learning the underlying relationship between tasks remains a challenge. Previous methods use a computational approach to identify task relationship by manually sampling feasible task relationships, training and evaluating the benefit of each sampled task relationship (Zamir et al., 2018; 2020). However, their scalability remains a serious issue as they require running the entire training pipeline for each candidate task relationship. (Bahl et al., 2023) adopts a multi-task structure for affordance. Compared with directly predicting affordance , the visual representation learning method is more flexible to fit various kinds of robot learning tasks with observation space. We advance multi-task learning by enabling the model to *automatically learn* the task relationship during training. Our method explicitly helps each task to learn to query useful information from other tasks.

## 3 METHODOLOGY

In recent advancements within the field of visual-motor control, there has been a growing emphasis on harnessing the remarkable generalization capabilities of machine learning models to develop unique representations for robot learning. As representatives, R3M (Nair et al., 2022) proposes a large vision-language alignment model based on ResNet (He et al., 2016) for behavior cloning; MVP (Xiao et al., 2022) leverages masked modeling on Vision Transformer (ViT) (Dosovitskiy et al., 2020) to extract useful visual representation for reinforcement learning; EgoVLP (Qinghong Lin et al., 2022) learns video representations upon a video transformer (Bain

et al., 2021). To leverage their successes, we proposed to cultivate better representations for robotic manipulation by fine-tuning these vision backbones with human-oriented guidance from diverse human action related tasks. In the following sections, we introduce our Task Fusion Decoder, which is a general-purpose decoder that can work with any existing encoder networks. We then detail its training for multi-task structure. For the human-oriented tasks selection, we leverage three mutually related tasks in the hand object interaction benchmark from the Ego4D dataset for joint training. We describe them as follows.

The object state change classification (OSCC) task is to classify if there is a state change in the video clip; the point-of-no-return temporal localization (PNR) task is to localize the keyframe with state change in the video clip; the state change object detection (SCOD) task is to localize the hand object positions during the interaction process.

## 3.1 TASK FUSION DECODER

Previous works primarily incorporate high-level information from the entire visual scene, often overlooking the vital influence of human motion within the representation. However, human knowledge such as hand-object interactions in the environments is important for robotic manipulations. To gather different human pre-knowledge concurrently, it is crucial to incorporate different temporal and spatial tasks simultaneously into a single representation. Also, different vision tasks should have information interaction, for the human-like synesthesia. To achieve this, we design a decoder-only network structure Task Fusion Decoder, which can both induce task-specific information and integrate different tasks.

Task Fusion Decoder is a multitask learner (see Figure 15) aiming to learn three human-oriented tasks which are originally from the ego-centric video dataset Ego4D (Grauman et al., 2022): object state change classification (OSCC), point-of-no-return temporal localization (PNR), and state change object detection (SCOD). The definition for the three tasks can be found in Figure 3. It is also designed to work with various vision backbones, such as ResNet (He et al., 2016), ViT (Dosovitskiy et al., 2020), and Timesformer (Bain et al., 2021). Given a video, we denote its number of input frames as $T$, the outputted number of patches (for ViT) or feature map size (for ResNet) per frame as $P$, and the representation dimension for the encoder as $D$. In this way, we can have: (1) the global feature $h_{cls} \in \mathbb{R}^{1 \times D}$ representing the whole video sequence, $e.g.$, the class token for ViT or final layer feature for ResNet; and (2) $h_{total} \in \mathbb{R}^{(P \times T) \times D}$ as dense features with spatial and temporal information preserved.

For time-related tasks, representation $h_t$ for the whole video sequence is required for learning. We choose $h_{cls}$ as $h_t$ and adopt a time positional embedding to localize the frame. For spatial-related tasks, representation $h_s$ for capturing the localization of one specific action, so we adopt a frame pre-selection strategy to select the keyframe that only covers the state change frame from $h_{total}$. In this case, $h_s \in \mathbb{R}^{P \times D}$ denotes the representation of the state change frame. Similarly, we adopt a positional encoding for $h_s$ before feeding into the decoder network. For ResNet, we append an additional transformer encoder network to adapt the convolutional feature to the patch-wise feature.

Within Task Fusion Decoder, we define 10 task tokens $z_i^k$ as the input of the $k_{th}$ decoder layer, where $1 \leq k \leq N$, $z_1^k$ and $z_2^k$ are object state change classification(OSCC) task token and temporal localization(PNR) task token, respectively; $z_3^k - z_{10}^k$ are state change object detection(SCOD) task tokens, which provide nominated bounding boxes for hand and object detection. The $k_{th}$ layer of the decoder structure can be formulated as:

$$\{f_i^k\}_i = \text{Self-Attention}(\{z_i^k\}_i) \tag{1}$$

$$\{z_i^{k+1}\}_i = \text{Cross-Attention}(h_t, \{f_i^k\}_i), i \in \{1, 2\} \tag{2}$$

$$\{z_i^{k+1}\}_i = \text{Cross-Attention}(h_s, \{f_i^k\}_i), 3 \leq i \leq 10 \tag{3}$$

where $f_i^k$ is the feature after interacting between task tokens, $z_i^{k+1}$ is the feature of next layer decoder input. Self-attention can perform task fusion for each layer. For the last layer of the decoder network, we adapt 10 MLP layers for 10 different task tokens as translators for the tasks with human pre-knowledge.

## 3.2 JOINT MULTITASK TRAINING

For the OSCC task, there is a binary label to represent whether a state changes or not. The decoder output is the probability of containing a state change in the input video sequence. The loss of OSCC task $L_{oscc}$ is thus a cross-entropy loss as a two-category classification problem.

For the PNR task, the label $D_{pnr}$ is a distribution with the length of number frames $T$, where the label of the state change frame is 1, and others are 0. For video clips without state change, all label is set to $1/T$. We mimic the assigned distribution with KL-divergence loss as follows:

$$L_{pnr} = \text{KL}(f(z_2^N), D_{pnr}) \qquad (4)$$

where $f(z_2^N)$ is the decoder output probability distribution for state change frame, while $D_{pnr}$ is the ground truth state change frame distribution.

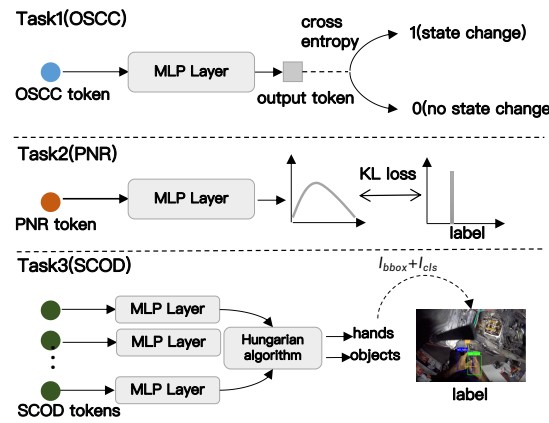

Figure 3: Definition for the three spatial and time related tasks and the formulation of the loss function.

For the SCOD task, we formulate it an object detection task following DETR (Carion et al., 2020), which uses the Hungarian algorithm (Kuhn, 1955) to select the most nominated bounding boxes for hands and objects. The decoder outputs are logits for bounding-box positions and object classes. We get the $L_{scod}$ by a bounding box localization loss and a classification loss.

For joint training of the three multi-tasks, we propose to balance the three losses by adding weighted terms as a variance constraint (Kendall et al., 2018) for them:

$$L = \frac{1}{2\sigma_1^2}L_{oscc} + \frac{1}{2\sigma_2^2}L_{pnr} + \frac{1}{2\sigma_3^2}L_{scod} + \log(\sigma_1\sigma_2\sigma_3), \qquad (5)$$

where $\sigma_i$ is a learnable variance. By leveraging such a constraint, the three tasks are automatically learned in a balanced manner.

## 4 EXPERIMENTS

### 4.1 IMPLEMENTATION DETAILS

We leverage our Task Fusion Decoder to finetune three backbone models that are frequently used in robotics tasks: R3M, MVP, and EgoVLP. The FHO slice of the Ego4D dataset is used. The

Table 1: Success rate evaluation on R3M model. We indicate performance decrease in Blue and performance increase in Red.

| env | | R3M (%) | R3M+ours (%) |
|---|---|---|---|
| kitchen | sdoor-open | 64.00 | **79.00** (+15.00) |
| | ldoor-open | **38.33** | 29.00 (-9.33) |
| | light-on | 75.00 | **77.34** (+2.34) |
| | micro-open | 27.34 | **28.67** (+1.33) |
| | knob-on | **61.34** | 58.00 (-3.34) |
| | average | 53.20 | **54.40** (+1.20) |
| metaworld | assembly | 93.67 | **98.67** (+5.00) |
| | bin-pick | 44.67 | **56.33** (+11.66) |
| | button-press | 56.34 | **62.67** (+6.33) |
| | hammer | **92.67** | 86.34 (-6.33) |
| | drawer-open | **100.00** | 100.00 (+0.00) |
| | average | 77.47 | **80.80** (+3.33) |
| adroit | pen | 67.33 | **70.00** (+2.67) |
| | relocate | 63.33 | **66.22** (+2.89) |
| | average | 65.33 | **68.11** (+2.78) |

Figure 4: Tasks defined in Kitchen, MetaWorld and Adroit environments from different views.

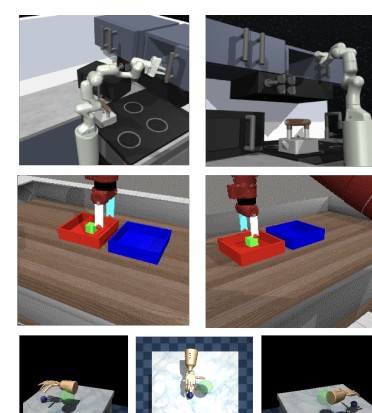

Table 2: Success rate evaluation on the EgoVLP and MVP models.

| env | | EgoVLP (%) | EgoVLP+ours (%) | MVP (%) | MVP+ours (%) |
|---|---|---|---|---|---|
| kitchen | sdoor-open | 43.00 | **44.00** (+1.00) | 32.00 | **44.00** (+12.00) |
| | ldoor-open | 4.00 | **7.00** (+3.00) | 9.00 | **11.00** (+2.00) |
| | light-on | **19.00** | 12.00 (-7.00) | **18.00** | 15.00 (-3.00) |
| | micro-open | 11.00 | **16.00** (+5.00) | 4.00 | **7.00** (+3.00) |
| | knob-on | 11.00 | **14.00** (+3.00) | **6.00** | 4.00 (-2.00) |
| | average | 17.60 | **18.60** (+1.00) | 13.80 | **16.20** (+2.40) |
| metaworld | assembly | 10.67 | **21.33** (+10.66) | 14.67 | **27.33** (+12.66) |
| | bin-pick | 4.67 | **12.00** (+7.33) | 3.33 | **4.00** (+0.67) |
| | button-press | **24.00** | 15.33 (-8.67) | **40.67** | 32.00 (-8.67) |
| | hammer | 58.00 | **81.33** (+23.33) | **98.67** | 97.33 (-1.34) |
| | drawer-open | 62.67 | **88.67** (+26.00) | 40.67 | **44.00** (+3.33) |
| | average | 32.00 | **43.73** (+11.73) | 39.60 | **40.93** (+1.33) |
| adroit | pen | 67.33 | **69.33** (+2.00) | 60.67 | **62.00** (+1.33) |
| | relocate | 26.67 | **32.00** (+5.33) | 16.00 | **19.33** (+3.33) |
| | average | 47.00 | **50.67** (+3.67) | 38.34 | **40.67** (+2.33) |

training dataset contains 41,000 video clips and the validation dataset contains 28,000 video clips. We randomly sample 16 frames from each video clip as the input. The image resolution is $224 \times 224$. We adopt the training code base in (Qinghong Lin et al., 2022). For all training experiments, we set the learning rate to $3 \times 10^{-5}$ and the batch size to 66. The training takes three days on 5 A6000 GPUs with AdamW optimizer used.

## 4.2 EXPERIMENTAL RESULTS IN SIMULATORS

In this section, we verify that our finetuning strategy yields representation that improves the robot's imitation learning ability compared with directly using pretrained backbones in three simulation environments: Franka Kitchen, MetaWorld, and Adroit, shown in Fig. 4. In Kitchen and MetaWorld, the state is the raw perceptual input's embedding produced by the visual representation model. In Adroit, the state contains the proprioceptive state of the robot along with the observation embedding.

For R3M (Nair et al., 2022), we follow its evaluation procedure (Nair et al., 2022) to test our representation under the behavior cloning setting. We train an actor policy that maps a state to robot action over a total of 20,000 steps with the standard action prediction loss. The number of demonstrations used for training imitation policies in the three environments is 50, 25, and 100, respectively. During the evaluation process, we evaluate the policy every 1000 training steps and report the three best evaluation results from different visual views. The results are shown in Tab. 1. For EgoVLP and MVP, the number of demonstrations used for training imitation policies in the three environments is 10, 50, and 100, respectively. We evaluate policy every 5000 training steps and report the best result from different visual views. The results are shown in Tab. 2.

From Tab. 1 and Tab. 2, we observe that our fine-tuning strategy improves the policy success rate compared to directly using the backbones, indicating our method can help capture human-oriented and important representation for manipulation tasks.

## 4.3 ABLATION STUDY

In this section, we evaluate the success rate results with ablations on temporal-related tasks and spatial-related tasks to understand the benefits of inducing perceptual skills in the model and the necessity of different perceptual skills for different tasks. We use R3M as the base model and re-implement the training on the model with only time-related tasks and the model with only spatial-related tasks. We select five environments from Franka Kitchen, MetaWorld, and Adroit.

As shown in Tab. 3, in most environments, robotics require both spatial and temporal perceptual skills to enhance the representation of observations. However, in several environments, only one perceptual skill is sufficient, and the other may have a negative effect. In the 'ldoor' environment, we believe that time information plays a leading role because capturing state changes over time can

Table 3: Ablation study about time-related tasks and spatial-related tasks.

| env | R3M | R3M+time | R3M+spatial | Ours(R3M+spatial+time) |
|---|---|---|---|---|
| micro | 23.00 | 25.00 | 26.00 | **28.00** |
| light | 67.00 | 75.00 | 70.00 | **83.00** |
| ldoor | 41.00 | **46.00** | 23.00 | 32.00 |
| assembly | 84.00 | 84.67 | 83.33 | **92.67** |
| relocate | 36.67 | 37.33 | **40.00** | 36.67 |

Table 4: The OSCC and PNR task results on the Ego4D benchmark.

| Model | Video-Text Pretrained | OSCC ACC% (↑) | PNR ERR (seconds) (↓) |
|---|---|---|---|
| TimeSformer | Imagenet Init. | 70.3 | 0.616 |
| TimeSformer | EgoVLP | 73.9 | 0.622 |
| Ours | EgoVLP | **76.3** | **0.616** |

be challenging. In the 'relocate' environment, spatial perception takes the lead as objects in the manipulation scene are readily apparent.

## 4.4 REAL-WORLD ROBOT EXPERIMENT

**Dataset.** We collect a Fanuc Manipulation dataset for robot behavior cloning, including 17 manipulation tasks and 450 expert demonstrations, as shown in Fig. 5. We employ a FANUC LRMate 200iD/7L robotic arm outfitted with an SMC gripper. The robot is manipulated using operational space velocity control. Demonstrations were collected via a human operator interface, which utilized a keyboard to control the robot's end effector. We established a set of seven key bindings to facilitate 3D translational, 3D rotational, and 1D gripper actions for robot control. During these demonstrations, we recorded camera images, robot joint angles, velocities, and expert actions.

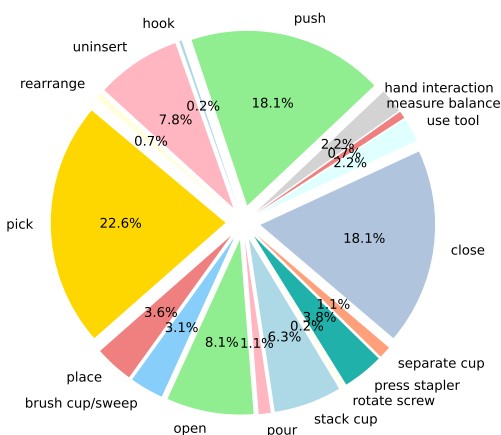

Figure 5: The distribution of our real-world robot dataset in a Fanuc robot, which covers many kinds of actions.

In the training phase of behavior cloning, we concatenate the robot's joint angles with encoded image features to form the input state. Rather than directly imitating expert actions in the robot's operational space (Nair et al., 2022), we opt to imitate the joint velocities derived from the collected joint trajectories. This approach allows for manipulation learning at a control frequency different from that of the human demonstrations, thereby offering flexibility in the network's inference time.

Fig. 6 presents experimental results for four representative tasks: pushing a box, closing a laptop, opening a drawer, and moving a cube to a specified location. During both training and evaluation, the robot arm's initial states and objects' initial states are randomized. We benchmark our approach against three existing methods: R3M, MVP, and EgoVLP. Our method outperforms most of these baselines across multiple tasks.

## 4.5 EVALUATION OF PERCEPTUAL TASKS ON EGO4D

To validate whether the multi-task network structure can capture task relationships and enhance computer vision representation, we employ our Task Fusion Decoder on the Ego4D Hand and Object Interactions benchmark. Due to label limitations, we re-implement our model using only time-related tasks, specifically OSCC and PNR. Subsequently, we evaluate the accuracy of object state change classification and temporal localization error in absolute seconds.

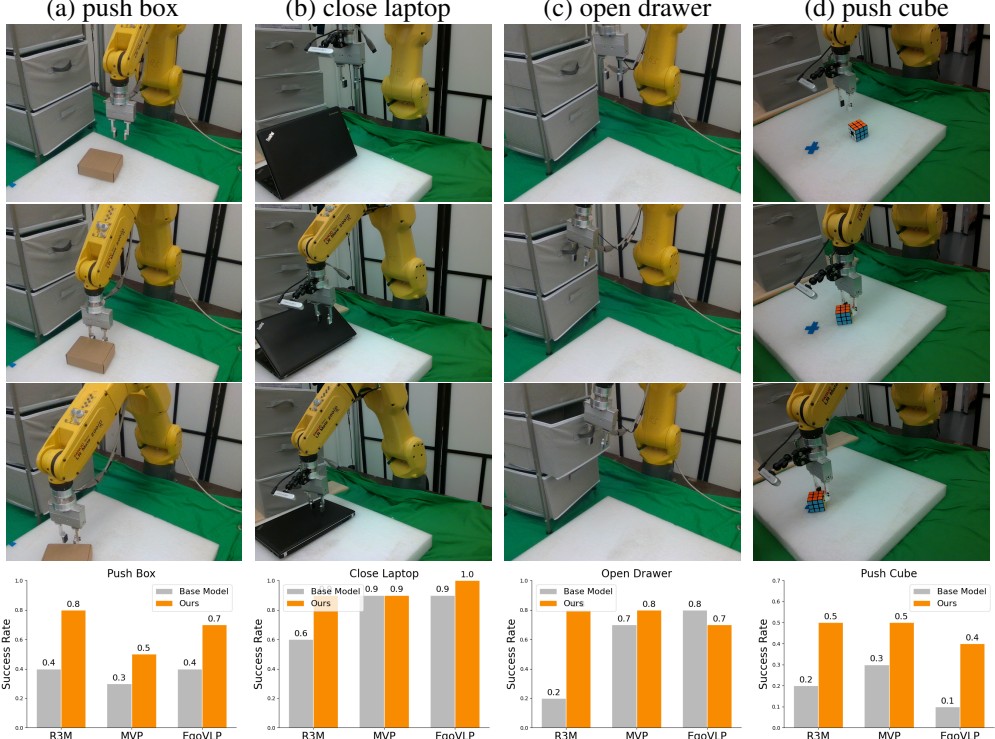

Figure 6: The result of our real robot experiments. The tasks are push the box, close the laptop, open the drawer, and push the cube from left to right.

From the results in Tab. 4, we observe that our model improves OSCC accuracy by 2.4% and reduces the PNR error by 0.006 seconds compared to the trained EgoVLP model. When compared to the ImageNet initialization model, our approach achieves a 6% improvement in OSCC accuracy while maintaining nearly identical PNR task performance. The strong result of these vision tasks verifies that our task fusion model can capture the task relationship hence making them benefit each other, showing effectiveness in learning a multi-task joint representation.

## 5 REPRESENTATION ANALYSIS

In this section, to demonstrate the effectiveness of our method, we first analyze the attention map in the manipulation scene to observe the impact of the spatial-related task. We then visualize the frame distribution at different times using a t-SNE figure (Van der Maaten & Hinton, 2008) to assess the effect of keyframe prediction.

### 5.1 ATTENTION MAP VISUALIZATION

The initial goal of the spatial-related task we designed is to capture the interaction between hands and manipulated objects and transfer it to the field of robotics manipulation. Therefore, we aim to demonstrate that our method places greater emphasis on the manipulation area while filtering out redundant information from the entire task area.

To validate our training strategy, we visualize the attention map of the last layer for R3M (ResNet) by Grad-CAM (Selvaraju et al., 2017). We separately visualize the attention maps for the original model, our fine-tuned model, and the ablative model, which includes only the time-related task, as shown in Fig. 7. We can see that: in both real robot scenes and simulation scenes, after the manipulation occurs, our method adjusts the representation to focus more on the action area, while the base model does not exhibit such an effect. Additionally, even with the time-related task, our method still cannot concentrate on the manipulation's local area, which confirms the effectiveness of the spatial-related task design in our network.

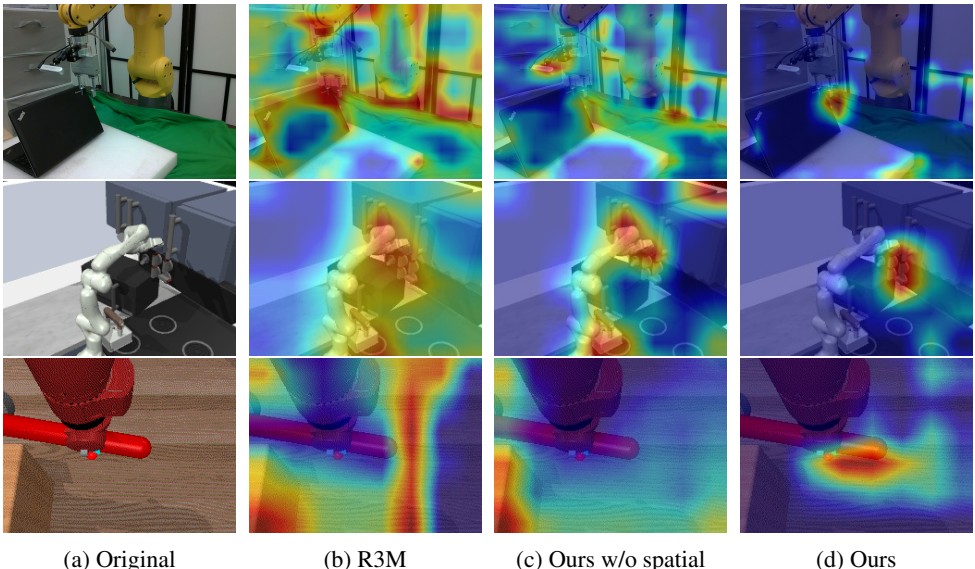

| (a) Original | (b) R3M | (c) Ours w/o spatial | (d) Ours |

Figure 7: The attention map visualization in different scenes. The pictures are original picture, base R3M model result and ours spatial ablation model result and ours model result from left to right.

## 5.2 T-SNE VISUALIZATION OF RRESENTATIONS

In this section, we plot the t-SNE figure for the representations of the whole sequence of the manipulation task in four kitchen environments at the same time. Because we add OSCC and PNR tasks for the human pre-knowledge for the model, which can capture the state change and predict the state change frame, the model will change the distribution for the representations of a manipulation task sequence.

As shown in Fig. 8, we classify each action sequence into before manipulation action and after manipulation

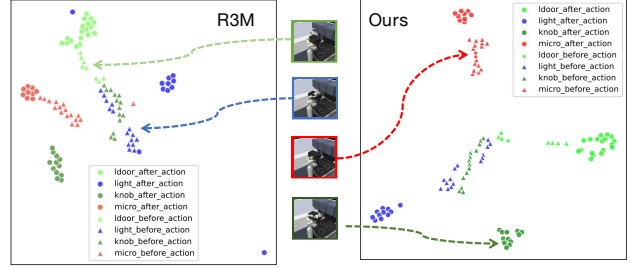

Figure 8: Left: the t-SNE figure for R3M model Right: the t-SNE figure for our model. Our model has a stronger ability to capture the state change

action. In more tasks, our model can have a bigger gap for representation in temporal, and get a clearer relationship between before-action and after-action representations.

## 6 CONCLUSION AND DISCUSSION

In conclusion, this work introduces a novel paradigm in the field of robot representation learning, emphasizing the importance of human-oriented perceptual skills for achieving robust and generalizable visual representations. By leveraging the simultaneous acquisition of multiple simple perceptual skills critical to human-environment interactions, we propose a plug-and-play module Task Fusion Decoder, which acts as an embedding translator, guiding representation learning towards encoding meaningful structures for robotic manipulation. We demonstrate its versatility by improving the representation of various state-of-the-art visual encoders across a wide range of robotic tasks, both in simulation and real-world environments. Furthermore, we introduce a real-world dataset with expert demonstrations to support our findings.

**Future work and broader impact.** In the future, we will explore the incorporation of a feedback loop or reward function into a joint visual representation learning and policy learning framework. Our approach has no ethical or societal issues on its own, except those inherited from robot learning.

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

# A APPENDIX

## A.1 NETWORK ARCHITECTURE AND CORRESPONDING TASK FUSION DECODER NETWORK DESIGN

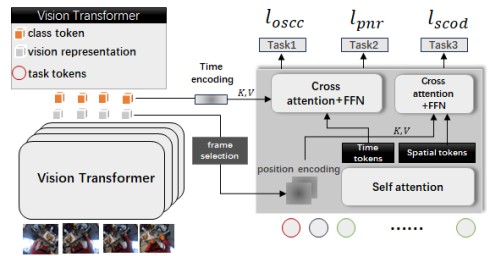 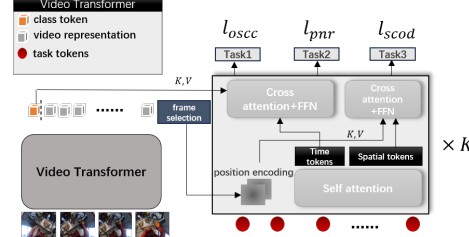

(a) Task fusion decoder for vision transformer.  (b) Task fusion decoder for video transformer.

Figure 9: Task fusion decoder for transformer structures.

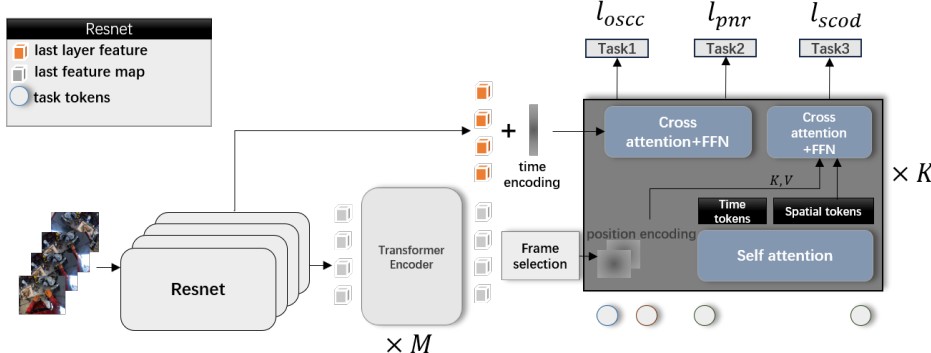

Figure 10: Task fusion Decoder for Resnet.

During the training process of Task Fusion Decoder, we need to set up different network structures for video transformer, vision transformer and Resnet. In this section, we introduce the details of the task fusion decoder in different networks.

**Video Transformer** encodes a video sequence into a single feature. One single video transformer encodes the video into class token feature $h_{cls} \in \mathbb{R}^{1 \times D}$ and total feature $h_{total} \in \mathbb{R}^{(T \times P) \times D}$. We directly use the class token feature $h_{cls}$ as our temporal feature $h_t$ for the embedding of the whole video sequence and adopt a frame-selector to get the keyframe feature $h_s \in \mathbb{R}^{P \times D}$ as shown in Fig. 9.

**Vision Transformer** encodes a single image into feature space, so we need $T$ encoder networks to deal with the $T$ frames input video separately. The class token representation $h_{cls} \in \mathbb{R}^{T \times D}$ is the gather representation for $T$ separate frames, and the total representation $h_{total} \in \mathbb{R}^{T \times P \times D}$. We also directly use $h_{cls}$ as our $h_t$ used for time-related decoder tasks. After that, we add a learnable time positional encoding to localize the frame. We also adopt the frame selection and add positional encoding for this model and get the spatial representation $h_s \in \mathbb{R}^{P \times D}$ as shown in Fig. 9.

**Convolution Based Network** like Resnet also encodes image representation separately. However, because there is a gap between convolution style representation and transformer style representation, we additionally add two layers transformer encoder for the representation to adapt the feature into a similar feature style as shown in Fig. 10.

## A.2 EXTENDED SIMULATION RESULTS

The result mentioned in the paper is the average result from different views, thus, we visualize the results from different views. As the results shown in Tab. 5 and Tab. 6, our method outperforms the base model in most views from different environments.

Table 5: Success rate evaluation on R3M model from different views in Kitchen and Metaworld environments.

| env | | Left camera | | Right camera | |
|---|---|---|---|---|---|
| | | R3M (%) | R3M+ours (%) | R3M (%) | R3M+ours (%) |
| kitchen | sdoor-open | 66.00 | **82.67** | 62.00 | **75.33** |
| | ldoor-open | **35.33** | **35.33** | **41.33** | 22.67 |
| | light-on | 70.67 | **76.00** | **79.33** | 78.67 |
| | micro-open | 36.00 | **38.67** | **18.67** | **18.67** |
| | knob-on | **66.67** | 62.00 | **56.00** | 54.00 |
| | average | 54.93 | **58.93** | **51.47** | 49.87 |
| metaworld | assembly | 94.67 | **100.00** | 92.67 | **97.33** |
| | bin-pick | 62.00 | **69.33** | 27.33 | **43.33** |
| | button-press | 62.00 | **64.00** | 50.67 | **61.33** |
| | hammer | **96.67** | 88.67 | **88.67** | 84.00 |
| | drawer-open | **100.00** | **100.00** | **100.00** | **100.00** |
| | average | 83.07 | **84.40** | 71.87 | **77.20** |

Table 6: Success rate evaluation on R3M model from different views in adroit environment.

| env | | View-1 | | Top-view | | View-4 | |
|---|---|---|---|---|---|---|---|
| | | R3M (%) | ours (%) | R3M (%) | ours (%) | R3M (%) | ours (%) |
| adroit | pen | 64.67 | **69.33** | **71.33** | **71.33** | **66.00** | **66.00** |
| | relocate | **69.33** | 67.33 | 62.00 | **70.67** | 58.67 | **60.67** |
| | average | 67.00 | **68.33** | 66.67 | **71.00** | 62.34 | **63.34** |

## A.3 REAL WORLD ROBOT EXTEND EXPERIMENT

In this section, we present visualizations of evaluation trajectories from real-world robot experiments, which illustrate the distinctions between our model and the base model. From top to bottom, we showcase both our success and failure results in comparison to the base model.

In the 'Opening Drawer' task, the base model frequently misses the target drawer handle, whereas our model succeeds in handling it. Similarly, in the 'Closing Laptop' task, the base model's robot arm often slides over the laptop's edge.

## A.4 TRAINING AND TESTING SETTING

For the finetuning of the large vision encoder, we freeze the first 2/3 layers of the vision encoder, and keep the last 1/3 layers of the vision encoder trainable. For example, for the EgoVLP pretrained model with a total of 12 layers, we maintain the last four layers trainable, which can both keep large model representation and induce human perceptual tasks for large vision encoder. During the deployment on robotics tasks, the vision encoder, whose representation has been shifted by the task fusion decoder, keep frozen and train a robotics policy.

## A.5 LEARNABLE SIGMA

In this section, we visualise the sigma learned during the multi-task training. As show in Fig. 16, Fig. 17, Fig. 18, sigma1 and sigma 3 shows decrease during the training process. The sigma for OSCC is about -1.3e-5, for PNR is about -3e-5, for SCOD is about 0.01, while the loss value of SCOD is about 1e3 times over PNR and OSCC.

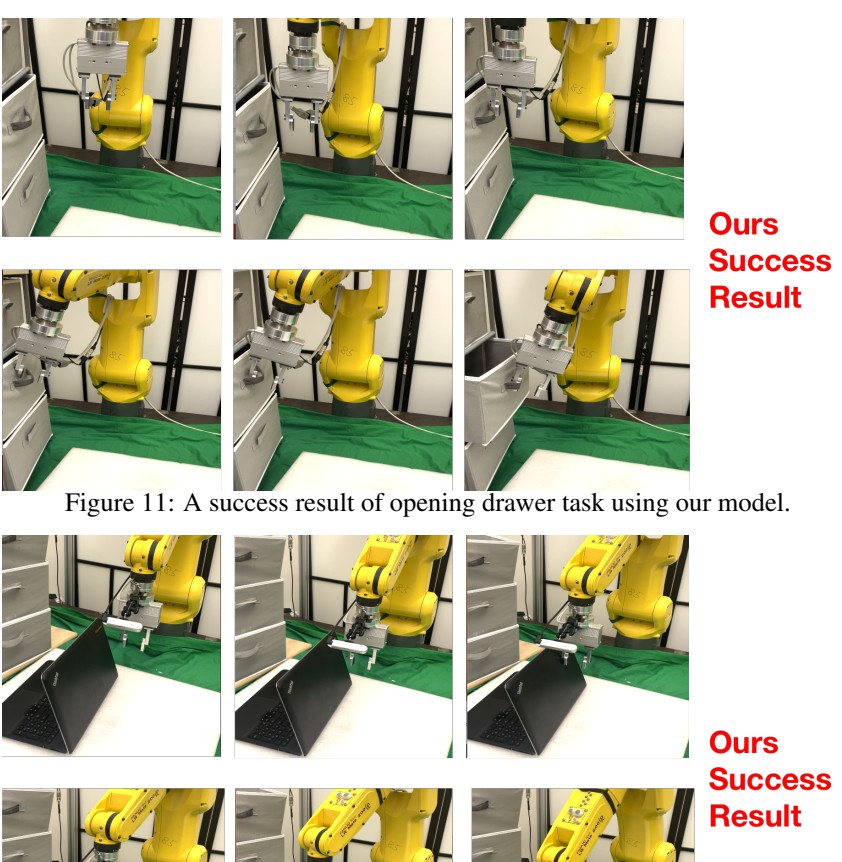

Figure 11: A success result of opening drawer task using our model.

Figure 12: A success result of closing laptop task using our model.

## A.6 ROBOT POLICIES DECODING

We decode our robot policy in this section for different tasks head. As shown in the figures below, most tasks meet the expection for the oscc, pnr and scod prediction. For robotics task, most bounding boxes can capture the contact position, which is from hand object interaction points.

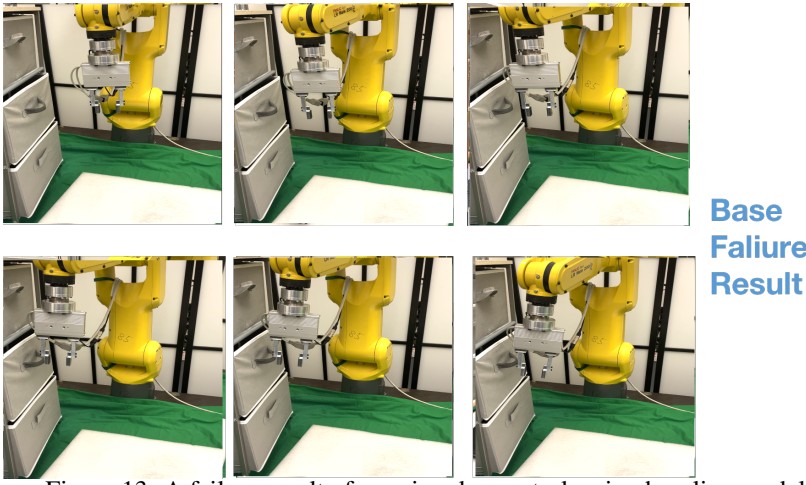

Figure 13: A failure result of opening drawer task using baseline model.

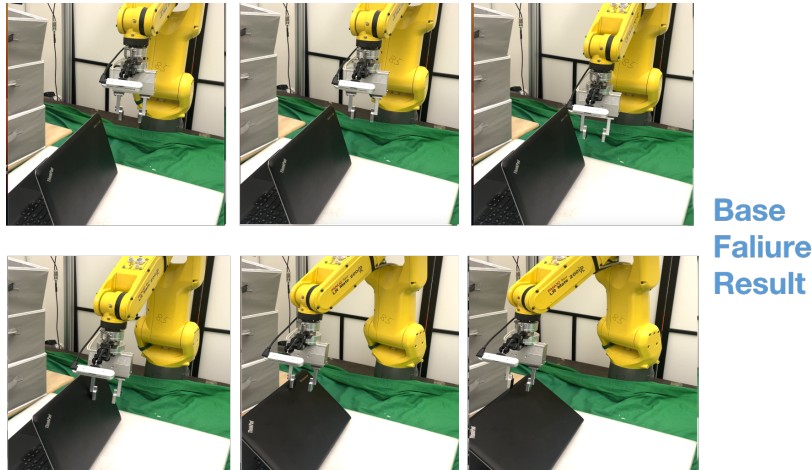

Figure 14: A failure result of closing laptop task using baseline model.

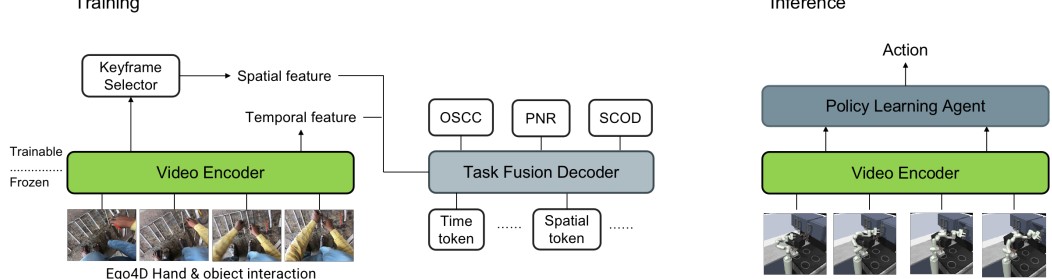

Figure 15: The training and inference setting in our work.

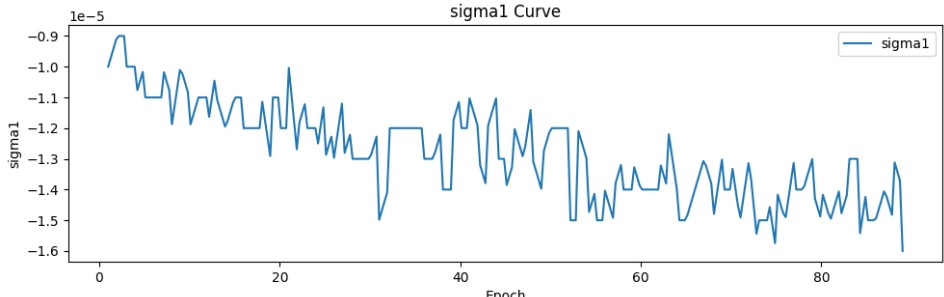

Figure 16: Sigma 1 training process.

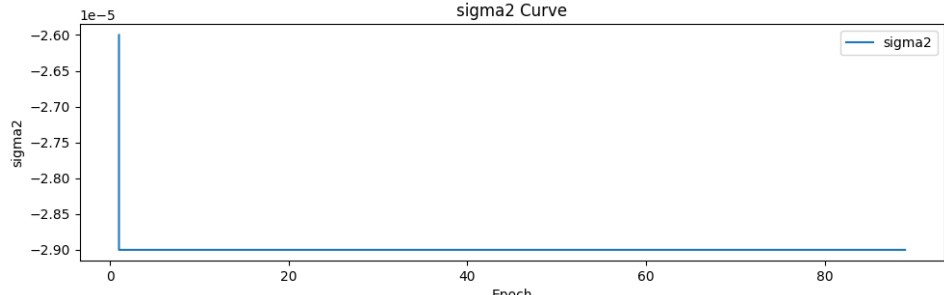

Figure 17: Sigma 2 training process.

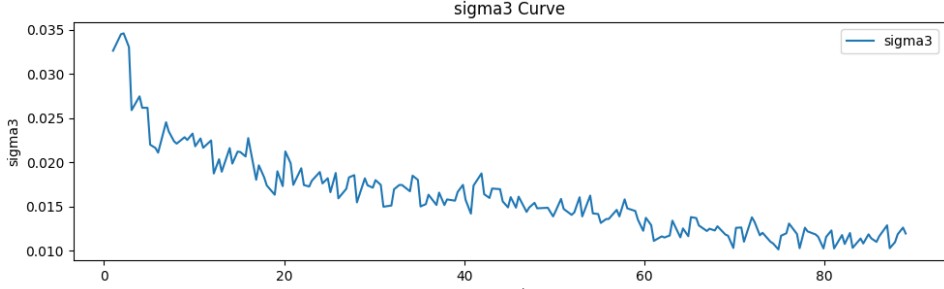

Figure 18: Sigma 3 training process.

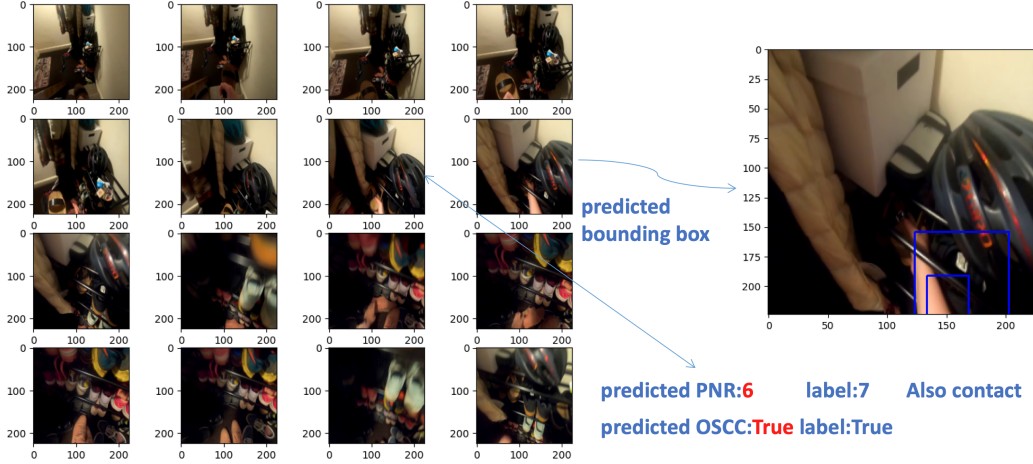

Figure 19: Human Policies Decoding – example 1.

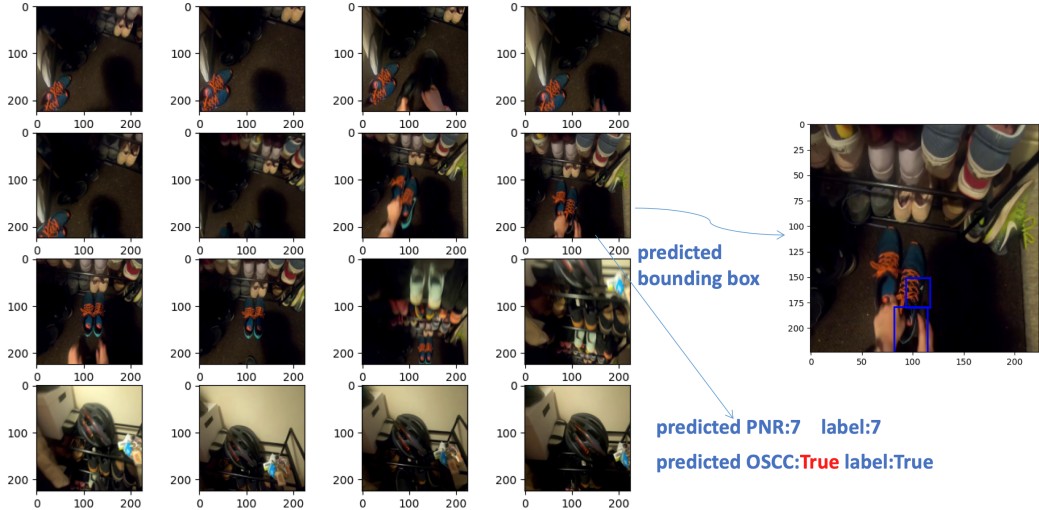

Figure 20: Human Policies Decoding – example 2.

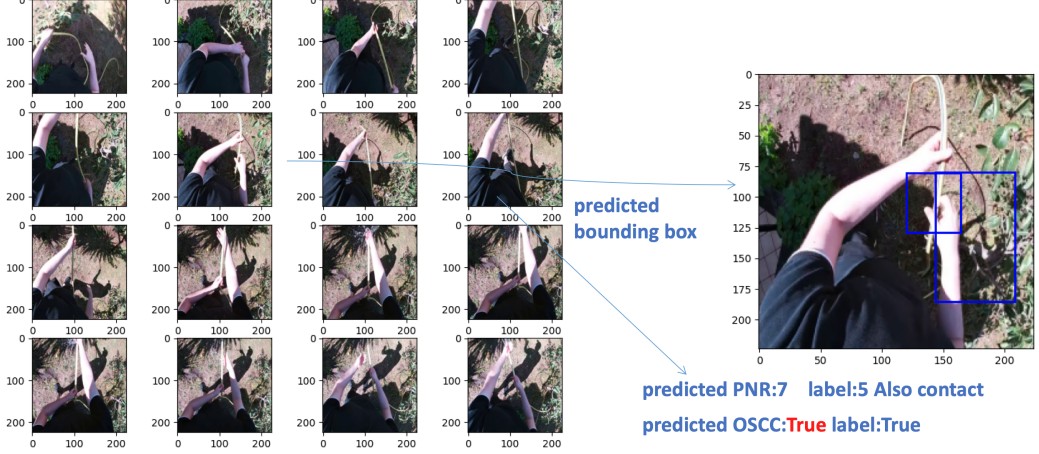

Figure 21: Human Policies Decoding – example 3.

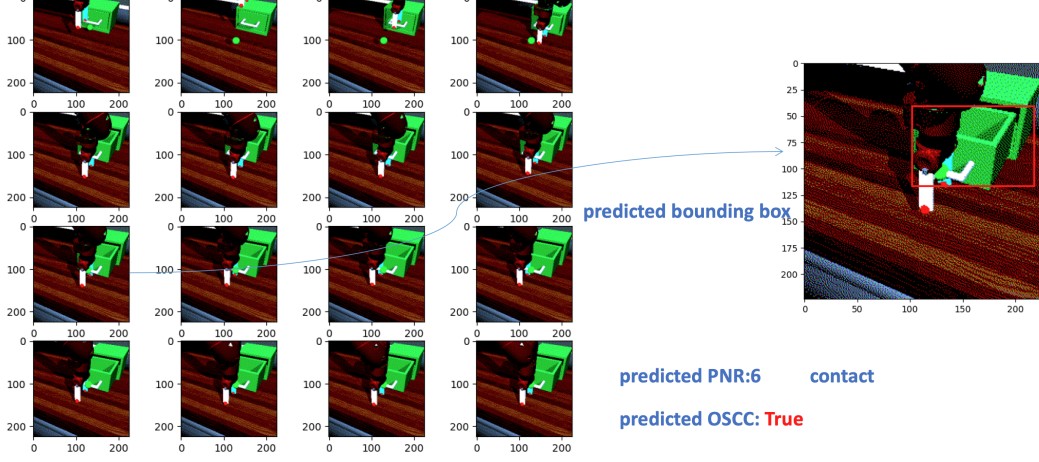

Figure 22: Robot Policies Decoding – example 4.

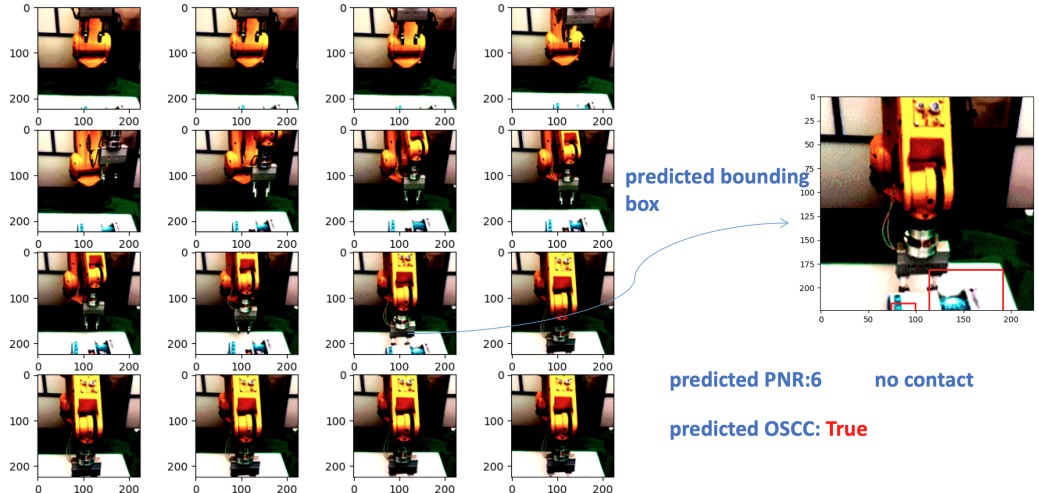

Figure 23: Robot Policies Decoding – example 5.

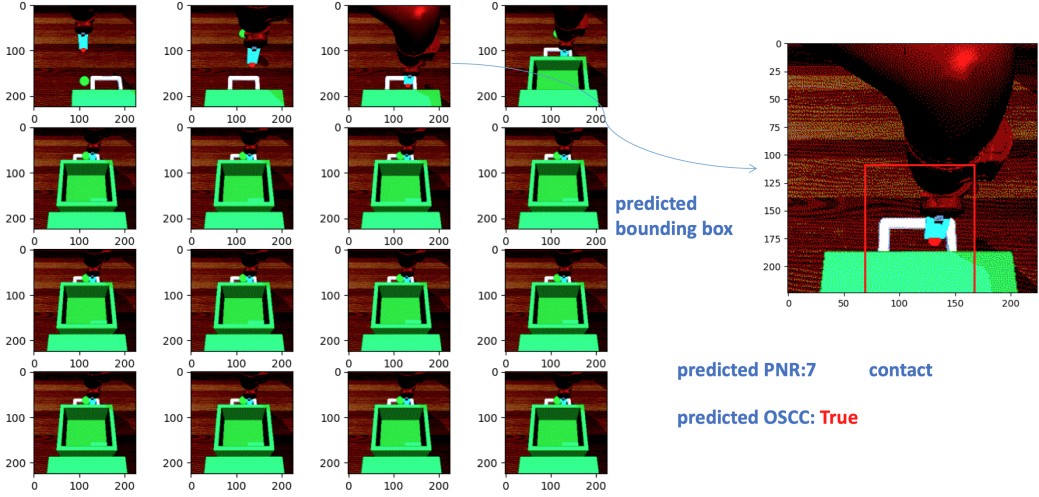

Figure 24: Robot Policies Decoding – example 6.

