# OpenReview forum: "Human-oriented Representation Learning for Robotic Manipulation"
_ICLR.cc/2024/Conference — Submitted to ICLR 2024_

### Official Review · Reviewer_kBXX · 2023-10-28

**Soundness:** 2 fair
**Presentation:** 2 fair
**Contribution:** 3 good
**Rating:** 6
**Confidence:** 3

**Summary:**

The paper introduces a human-centric multi-task decoder that extracts varied task information from Ego4D videos. This learnt representation is subsequently utilized to refine pre-trained visual encoders. The results demonstrate that their novel task fusion decoder consistently enhances the performance of three leading visual encoders, namely R3M, MVP, and EgoVLP, for subsequent manipulation policy learning.

**Strengths:**

The paper presents an innovative approach to maximize the extraction of hidden information from human tasks in Ego4D videos. This information is then employed as an initialization for refining premier pre-trained encoders for robotic manipulation. The study offers a comprehensive set of cross-simulator experiments and real-world domain tests, underscoring the importance of these human-oriented features from Ego4D for pre-training robotic encoders.

**Weaknesses:**

1. Recent progress in VLM has showcased vast capabilities in reasoning about state-changes, object detection, and identifying irreversible states. Could these representations or features from a pre-trained VLM be harnessed to refine R3M, MVP, or EgoVLP, rather than the method proposed?

2. An observation suggests that tasks across various environments and models tend to underperform in toggle on/off tasks. Is there an underlying reason for this trend?

3. The ablation studies indicate that both spatial and temporal tasks, when treated independently, don't yield impressive results. However, their combination surpasses the baseline. Can the authors provide an intuition behind this phenomenon?

4. Figure 5 could benefit from a reduction in font size for the percentage scores. In Figure 6, it might be more efficient to omit repetitive legends across plots.

5. The authors might consider leveraging other task or robot demonstration datasets featuring a third-person perspective with actual robot embodiment beyond Ego4D. A few notable examples include:
    - Padalkar et al. "Open X-Embodiment: Robotic learning datasets and RT-X models." arXiv:2310.08864 (2023).
    - Kumar et al. "RoboHive: A Unified Framework for Robot Learning." arXiv:2310.06828 (2023).
    - Duan et al. "AR2-D2: Training a Robot Without a Robot." arXiv:2306.13818 (2023).
    - Chen et al. "Learning generalizable robotic reward functions from 'in-the-wild' human videos." arXiv:2103.16817 (2021).

6. Have the authors evaluated their method on tasks with longer horizons or those more intricate than mere simple action primitives?

**Questions:**

All my questions and doubts are listed in the weakness section, i hope the author could address them.

---

> ### Author Response · Authors · 2023-11-23
> **Response to Reviewer kBXX**
>
> We would like to thank the reviewer for the positive feedback and constructive comments. We are glad that the reviewer found our approach innovative.
>
> **Q1. [Recent progress in VLM has showcased vast capabilities in reasoning about state changes, object detection, and identifying irreversible states. Could these representations or features from a pre-trained VLM be harnessed to refine R3M, MVP, or EgoVLP, rather than the method proposed?]** \
> We agree that this is a very interesting suggestion! Indeed, the recent advances in VLM have demonstrated a strong representation of learning ability. One potential approach to harnessing the representation from VLM is via distillation where we distill the foundational VLM’s representation extraction ability into smaller models such as R3M. We are very excited about investigating this open problem in our future work!
>
>
> **Q2. [An observation suggests that tasks across various environments and models tend to underperform in toggle on/off tasks. Is there an underlying reason for this trend?]** \
> We follow DETR for the finetuning of our network. In the object detection area, the DETR is more fit for large object detection. We hypothesize the toggle is too small, so our representation is not sensitive to that.
>
>
> **Q3. [The ablation studies indicate that both spatial and temporal tasks, when treated independently, don't yield impressive results. However, their combination surpasses the baseline. Can the authors provide an intuition behind this phenomenon?]** \
> We hypothesize this is attributed to our task fusion decoder, which explicitly builds feedback paths between tasks and helps each task to learn to query useful information from other tasks, thus making tasks interact with and complement each other, ultimately helping the encoder learn more useful representations compared to learning individual task.
>
>
> **Q4. [Figure 5 could benefit from a reduction in font size for the percentage scores. In Figure 6, it might be more efficient to omit repetitive legends across plots.]** \
> We would like to thank the reviewer for this great suggestion! We will incorporate your suggestions in the final paper version.
>
>
> **Q5. [The authors might consider leveraging other task or robot demonstration datasets featuring a third-person perspective with actual robot embodiment beyond Ego4D. A few notable examples include Open X-Embodiment, RoboHive, AR2-D2, etc.]**  \
> We would like to thank the reviewer for providing these related papers and datasets! We are excited about leveraging these new datasets to test our representation learning approach in our future work!
>
>
> **Q6. [Have the authors evaluated their method on tasks with longer horizons or those more intricate than mere simple action primitives?]** \
> We would like to thank the reviewer for this great suggestion! In this paper, we consider tasks with relatively short horizons, so the proposed method is fairly compared with baselines. We are excited about evaluating our learned representation for long-horizon tasks in our future work!
>
> \
> Thanks again for your time and effort! For any other questions, please feel free to let us know.

---

### Official Review · Reviewer_UxCD · 2023-10-30

**Soundness:** 2 fair
**Presentation:** 3 good
**Contribution:** 3 good
**Rating:** 5
**Confidence:** 4

**Summary:**

This paper introduces a task-fusion decoder architecture that fine-tunes representations for robot manipulation tasks using ego-centric video and semantic information about tasks and task stages present in these. The key idea is to us supervision around object state changes (when and if this occurred) and hand/object presence to improve representations, with the hypothesis that representations learned by mastering multiple skills allow for improved robot manipulation. Experiments are conducted to fine-tune a number of back-bone models using the Ego4D dataset, and then measuring their behaviour cloning task success performance on a number of simulated benchmarks )Kitchen, MetaWorld, Adroit) in addition to a real world setting with a Fanuc arm. Results appear to show improved success rates across most tasks evaluated, and ablations indicate that both spatial (object localisation) and temporal information (when state changes occur) contribute to this effect. In general the idea is clear, although the paper lacks some clarity around the methodology that inhibits the core ideas getting across.

**Strengths:**

* The proposed approach looks to be very flexible, and easily added to existing backbones.
* The proposed approach introduces a useful means of exploiting more readily available data of humans performing tasks in robot manipulation tasks where data is typically not as easily obtained, without the need for direct correspondences between human and robot actions or behaviours.

**Weaknesses:**

My primary concerns with the paper are around clarity, and the motivation for the choice of auxiliary task decoder objectives.

Major:

As I understand it, the proposed approach learns to decode 3 aspects related to embodied tasks from video/image representations, object state change classification, point of no return temporal localisation and state change object detection, using self attention to learn relationships between tasks, and cross attention to decode tasks. However, the paper does not have a clear definition of these different objectives, and I found this relatively hazy throughout. I think the work would be much clearer if this was defined much earlier (eg. pg 4). It may also be worth rethinking the acronyms or descriptions used here, since these terms (OSCC,PNR,SCOD) do not seem particularly explanatory, and raise more questions to the reader. For example, it is not apparent to me what the difference between state change object recognition and object state change classification is, or what the point of no return is. These need to be more crisply defined and explained earlier. The text notes that these are defined in Fig 3, but fig 3 contains no definition, just a set of loss functions and model visualisations. The associated descriptions on page 5 are also confusing to me, and raise more questions (see below) than answers.

Fig 2 shows a nice depiction of the decoder, and has helpful labelling to link to equations. It would be helpful for the global and temporal features $(h_*)$ to be represented on here too, and for a functional description of these operations to be provided. As it stands, I am unclear whether the + in the figure denotes concatentation, attention or something else. The appendix figures seem to indicate a transformer look-up, but this would be a lot clearer if these figures were more explicitly mapped to the nomenclature in the text.

It is unclear to me why the particular tasks to be decoded were selected, and why this one should be used over various other options or formulations exploiting the human data. The task losses seem somewhat arbitrary to me. This is motivated from a cognitive science perspective (Kirkham et al., 2002), but this aspect could be strengthened to make this choice seem less heuristic.

Results seem to be strong, but no indication as to the significance of these is provided (eg. error bars etc.), so this is difficult to evaluate.

Qualitative results in 5.1 are not very informative (t-sne produces different results when re-run), so unclear if this is just a projection effect. Similarly, attention maps have known issues. I think a stronger piece of evidence would be to decode the robot policies and see if these actually map to reasonable event changes, and object/hand bounding boxes.

Minor:

Training takes three days on 5 A6000 GPUs - this seems to be a very expensive fine-tuning operation for the performance gains observed. I recognise that the convention is to evaluate these tasks in terms of success rate, but it may be time to think in terms of more than success and evaluate performance as a function of compute.

**Questions:**

The decoder outputs information about state changes. Could you define what constitutes a state change, or what the state is in these tasks? This knowledge or choice seems to explicitly impose structure on the type of control task being considered.

The task fusion decoder uses 10 tokens? Why was this number chosen?

The point of no return is defined as the task to localise the keyframe with state change in the video clip. Where does the "no-return" terminology come from and what does this mean?

Can you clarify what the distributions are in the KL loss function in (4)?  These aren't categorical, since I assume there can be multiple state changes in a sequence... How is this loss actually computed?

Why are task 1 and task 2 different? It seems to me that predicting when a state change occurs already includes the task of predicting if a state change occurred.

The SCOD task is defined as localisation the hand object positions during the interaction process, which seems to apply continuously, but the terminology state change object detection implies this is only when a state change occurs. Which is is?

I appreciate the novelty in the proposed task decoding approach, but is it an overreach to claim that this paper represents a *paradigm shift* in representation learning? The idea of learning representations by exploiting human data is relatively well established eg. Sermanet et al. Time-Contrastive Networks: Self-Supervised Learning from Video 2018.

Table 1. Please provide more detail on this - are success rates averaged over multiple repetitions (if so, how many?) Are there error bars, are these results significant?

Will the dataset in Fig 5. be released fro reproducibility?

---

> ### Author Response · Authors · 2023-11-23
> **Response to Reviewer UxCD(Part 1/2)**
>
> Thank you for your efforts to improve the quality of our papers. We thank you for pointing out your concerns, which we believe have been carefully clarified and addressed, as follows.
>
>
> **Q1. [The decoder outputs information about state changes. Could you define what constitutes a state change, or what the state is in these tasks?]** \
> The state change label comes from the Ego4D hand-object interaction benchmark label, in the dataset, the state change moment is mostly defined by a contact between hand and object.
>
> **Q2. [The task fusion decoder uses 10 tokens. Why was this number chosen?]** \
> Yes, TFD uses 10 tokens for 3 tasks. The number of the tokens is determined by the tasks that we considered: the first token is for the OSCC task, the second token is for the PNR task, and the last eight tokens are for the SCOD task, which needs to generate 8 nominated bounding boxes for hands and objects.
>
>
> **Q3. [The point of no return is defined as the task to localize the keyframe with state change in the video clip. Where does the "no-return" terminology come from and what does this mean?]** \
> We apologize for the confusion. The no-return concept is also from the Ego4D dataset [A], it means that you now have to continue with what you are doing and it is too late to stop.
>
> [A] Kristen Grauman, Andrew Westbury, Eugene Byrne, Zachary Chavis, Antonino Furnari, Rohit Girdhar, Jackson Hamburger, Hao Jiang, Miao Liu, Xingyu Liu, et al. Ego4d: Around the world in 3,000 hours of egocentric video. In Proceedings of the IEEE/CVF Conference on Computer Vision and Pattern Recognition, pp. 18995–19012, 2022.
>
>
> **Q4. [Can you clarify what the distributions are in the KL loss function in (4)? These aren't categorical, since I assume there can be multiple state changes in a sequence... How is this loss actually computed?]** \
> Sorry for the confusion. The distribution is state change probability distribution, the decoder outputs a tensor with state change probability in frame dimension. The label is a tensor consisting of 0 and 1, indicating whether the state changes or not in the frame dimension.
> We treat the two tensors as distributions and measure the KL distance between them.
>
>
> **Q5. [Why are Task 1 and Task 2 different? It seems to me that predicting when a state change occurs already includes the task of predicting if a state change occurred.]** \
> We agree the two tasks have some similarities. However, they play different roles in our framework: the first indicates whether a state changes in the whole video sequence, and the second indicates when the hand contact with the object. The first one focuses on the whole video sequence, while the second focuses more on one moment.
>
>
>
> **Q6. [The SCOD task is defined as the localization of the hand object positions during the interaction process, which seems to apply continuously, but the terminology state change object detection implies this is only when a state change occurs. Which is it?]** \
> We apologize for the confusion. The SCOD is the task of detecting the object undergoing a state change from the given video clip.
>
> **Q7. [I appreciate the novelty in the proposed task decoding approach, but is it an overreach to claim that this paper represents a paradigm shift in representation learning? The idea of learning representations by exploiting human data is relatively well established eg. Sermanet et al. Time-Contrastive Networks: Self-Supervised Learning from Video 2018.]** \
> Thank you for the suggestion. We have removed the overclaimed expression.
>
> **Q8. [Table 1. Please provide more detail on this - are success rates averaged over multiple repetitions (if so, how many?) Are there error bars, are these results significant?]** \
> Currently, our experiment in Table 1, is evaluated 20 times and averages the three best successful rate results for every single environment. Also, for the kitchen and metaworld environments, the result is the average from two camera views, for the adroit environment, the result is the average from three camera views.  Thus, in a total of 12 environments, there are 460 evaluations.
>
> We do have error bars based on different views as the Table below:
> | env              |     R3M          | R3M+ours|
> | :--------------------------------------------- | :-----: | :--------: |
> | assembly | 93.67(3.77)|  **98.67**(1.89)  |
> | bin | 44.67(8.01)|  **56.33**(1.89)  |
> | hammer | **92.67**(5.66)|  86.34(3.30)  |
> | pen | 67.73(3.52)|  **70.00**(2.69)  |
> | relocate | 63.33(5.45)|  **66.22**(5.09)  |

---

> > ### Author Response · Authors · 2023-11-23
> > **Response to Reviewer UxCD(Part 2/2)**
> >
> > **Q9. [Will the dataset in Fig 5. be released for reproducibility?]** \
> > Of course. We open-source the dataset at [data_link](https://www.dropbox.com/scl/fi/tw7m6i7cgapefap7eozb3/fanuc_manipulation.zip?rlkey=53lz46vopb5r8r9l2d6d3jegt&dl=0), code at [code_link](https://www.dropbox.com/scl/fi/bdny032hk9q5jmed4doln/human_oriented_code.zip?rlkey=d8g3uvqr8t3s1kyz8edn7s8t1&dl=0).
> >
> > **W1. [The unclear for expressions and figures]** \
> > We appreciate your valuable suggestion for our expressions and figures. We have followed your advice and redrawn the Fig. 2 with a clearer label. We removed the “+” in the new version of Fig. 2, the spatial feature and temporal feature are two separate features fed into the task fusion decoder. Also, we reorganized the part about the three ego4d tasks and moved it to the front paragraph following your advice for a clearer understanding. We have added more explanations for the three tasks in Section 3.2.
> >
> > **W2. [It is unclear to me why the particular tasks to be decoded were selected, and why this one should be used over various other options or formulations exploiting the human data.]** \
> > In this paper, we selected three tasks from the ego4d dataset that are most related to human and object interaction and are critical for everyday scenarios as we expect that representations learned from these tasks could benefit the manipulation tasks. We would like to note that our multi-task framework is flexible and can be adapted to other tasks straightforwardly. We are excited about exploring other tasks in our future work.
> >
> > **W3. [I think a stronger piece of evidence would be to decode the robot policies and see if these actually map to reasonable event changes, and object/hand bounding boxes.]** \
> > We appreciate your suggestion for decoding the robot policies. We have added experiments in Appendix A.6 both for robot and human policies, most bounding box and keyframe predictions meet our expectation.
> >
> > **W4. [Training takes three days on 5 A6000 GPUs ]** \
> > The training time is constrained by the backbone and the dataset. Our method is also compatible with lightweight models that require less data for training. Improving training efficiency would be an exciting future work.
> >
> > \
> > We wish that our response has addressed your concerns, and turns your assessment to the positive side. If you have any questions, please feel free to let us know during the rebuttal window. We appreciate your suggestions and comments!

---

### Official Review · Reviewer_hzuR · 2023-10-31

**Soundness:** 3 good
**Presentation:** 4 excellent
**Contribution:** 3 good
**Rating:** 6
**Confidence:** 4

**Summary:**

This paper presents a multi-task learning framework for fine-tuning pre-trained visual deep representations on data relating to human-object interaction from the Ego4D dataset. The tasks used for fine-tuning are object state change classification (OSCC), point-of-no-return temporal localization (PNR), and state change object detection (SCOD). OSCC is performed based on entire video clips, while PNR detects at which particular frame in the video the state change happened, and SCOD applies to each frame of the video to localize the hand and objects that the hand is interacting with. The authors fine-tune a transformer which maps from the videos to the task outputs on top of three pre-trained, large visual models (R3M, MVP and EgoVLP). They show that the multi-task fine-tuning generally improves downstream success rates for agents interacting with three environments: MetaWorld, Adroit, Kitchen, and a selection of real-world robot tasks like opening a drawer or pushing a cube.

**Strengths:**

This paper tackles an important problem: how to adapt large pre-trained models to be useful for downstream tasks such as robotics. This is an increasingly important area as foundation models become more common, but such models can be too expensive to train for many research groups. Therefore, having methods that successfully adapt these models with less computationally expensive training is important. The main finding, that fine-tuning video models on specific tasks relating to hand / object interactions, is surprising given that this is the same data that the video models were trained on, just with a different objective. I believe the community will find this interesting.

The paper is thorough in its evaluation of the multi-task transformer (referred to as the “task fusion decoder”) fine-tuning. The paper covers three different pre-trained vision models as backbones, and three different downstream robotics environments for testing. While the fine-tuning does not universally help the performance in downstream tasks, it does improve the performance on average, and in some cases this improvement can be very significant (up to ~25% in success rate).

The paper is clearly written and easy to follow. Figures 2 and 3 are especially well constructed. They depict the model with sufficient clarity that it is easy to understand both the basic structure of the transformer, as well as the specific loss functions for each of the different tasks considered. The tables are well constructed, with helpful color coding to determine where the fine-tuning helps and where it does not. The methodology was clearly explained for how downstream tasks were evaluated. My only high-level figure recommendation would be to remove Figure 5, as the paper does not evaluate across all the tasks in the FANUC dataset, and so understanding the variety of actions within this dataset is not particularly important for the reader. It could easily be moved to the supplement.

**Weaknesses:**

There are two weaknesses to the paper that I believe can be addressed in rebuttals.

First, while the body of the paper is well written and easy to follow, the introduction is inappropriate given the results shown. The claimed message, that the presented fine-tuning represents a method for human alignment without requiring human labels, is not supported by the paper’s results and experiments. What the paper shows is that fine-tuning video models on hand-object interaction tasks improves the representation for robotics. This is absolutely not the same as having a scalable proxy for human evaluation. To make that claim, one would need to run a study comparing representations fine-tuned with human feedback to those fine-tuned with these hand-object interaction tasks.

Instead, I think what will interest the broader community is the fact that hand-object interactions are an exceptionally good choice of task for fine-tuning video models *even when exactly the same input data is used*. I was surprised that the base video models were also trained on Ego4D. Since the fine-tuning is also done on Ego4D, this means that the only difference is using the hand-object task for fine-tuning. But this simple change results in dramatic improvements on downstream success! This is critical for the community to know, since there are many possible tasks that could be chosen for fine-tuning.

To further underscore this point, another ablation that separately removes each of the different task heads would be very valuable. While there are space vs. time ablations in the current paper, there are three different task heads, and it would be nice to see the effect of each one on the downstream performance. Similarly, the loss uses values of “sigma” for each task contribution, and it would be great to see what the final learned values of these sigmas are (since that will be a proxy for how much the model is using the different task outputs for fine-tuning).

This is the main, original point of the paper. The presented task fusion decoder is not particularly original, there are many multi-task learning architectures involving transformers with different task heads (e.g. Bhattacharjee et al, CVPR 2022, MulT: An End-to-End Multitask Learning Transformer). Since this particular architectural choice is not the main message of the paper, it is not necessary to include comparisons to such architectures.

Second, and related to the point above, the paper is missing a key piece of related work for the main claim that hand-object interactions are an excellent task for fine-tuning models. The paper is Bahl et al, CVPR:2023, Affordances from Human Videos as a Versatile Representation for Robotics. Rather than fine-tuning a video model, Bahl et al train a model to predict hand-object interaction locations directly, and find that this supports many downstream robotics tasks. It would be valuable to know why one would prefer a fine-tuned video model on these interaction tasks, as opposed to directly predicting them from the outset.

**Questions:**

* Why would we expect fine-tuning on the task not to improve the performance on the task? (In relation to section 4.5)
* What about ablating each loss type separately?
* “Also, different vision tasks should have information interaction, for the human-like synesthesia.”
  * This does not really add anything. What is human-like synesthesia?
* For the real world experiments: how many test interactions were performed?
* Can the authors commit to open-sourcing the real-world dataset?
* “This is analogous to humans who, in specific environments, focus solely on one particular perception.”
  * Please remove this. Humans do not, in specific environments, focus solely on one particular perception. If they do, I would like to see the citation backing this up.
* Learned sigmas: what are they after training? When you say they are learned, is that per scene, or just differentiable but global?
* Please provide ablations for each individual task head

---

> ### Author Response · Authors · 2023-11-23
> **Response to Reviewer hzuR (Part 1/2)**
>
> Thank you for the insightful comments and suggestions.
>
> **Q1. [Why would we expect fine-tuning on the task not to improve the performance on the task? (In relation to section 4.5)]** \
> Sorry for the confusion. For Section 4.5 and Table 4, the first two rows' results are from ego4d pretraining and finetuning on specific results, and our results also pretraining from ego4d, but finetuning on two tasks at the same time with task fusion decoder. The results can, in some sense, show our tasks in the task fusion decoder have information exchange, which can benefit each one.
>
>
> **Q2. [What about ablating each loss type separately? Please provide ablations for each individual task head]** \
> Good question. We have added experiments by each task head, by retraining the model with three single tasks and retraining the policy for different models. We observed that in most situations, adding all three tasks gets the best result.
>
> |env                     | R3M+OSCC | R3M+PNR | R3M+SCOD | R3M+OSCC+PNR+SCOD|
> | :--------------------------------------------- | :-----: | :------------: | :---------: | :---------: |
> |micro | 21.00 |**28.00**|26.00|**28.00**|
> | light  | 82.00 | 69.00 | 70.00 | **83.00** |
> | ldoor| **39.00** |34.00 | 23.00 | 32.00 |
> | assembly|84.67|81.33| 83.33| **92.67** |
> | relocate| 39.33| **40.00**| **40.00**| 36.67 |
>
>
> **Q3.[“Also, different vision tasks should have information interaction, for the human-like synesthesia.” This does not really add anything. What is human-like synesthesia?]**\
> Human-like synesthesia is the way humans perceive spatial, temporal, and other information about the world simultaneously. In a task trajectory, humans gather information at the same time, like object localization, time to happen contact.
>
>
> **Q4. [For the real world experiments: how many test interactions were performed?]** \
> We perform 10 interactions for every single model in every single task following r3m evaluation, so in every task, there are 3 baseline models and 3 our models, a total of 240 interactions for Fig 5.
>
>
> **Q5. [Can the authors commit to open-sourcing the real-world dataset?]** \
> Sure. We open-source the dataset at [data_link](https://www.dropbox.com/scl/fi/tw7m6i7cgapefap7eozb3/fanuc_manipulation.zip?rlkey=53lz46vopb5r8r9l2d6d3jegt&dl=0), code at [code_link](https://www.dropbox.com/scl/fi/bdny032hk9q5jmed4doln/human_oriented_code.zip?rlkey=d8g3uvqr8t3s1kyz8edn7s8t1&dl=0).
>
> **Q6. [This is analogous to humans who, in specific environments, focus solely on one particular perception.”
> Please remove this. Humans do not, in specific environments, focus solely on one particular perception. If they do, I would like to see the citation backing this up.]** \
> Thank you for your advice. We have removed this in our paper.
>
> **Q7. [Learned sigmas: what are they after training? When you say they are learned, is that per scene, or just differentiable but global?]** \
> We retrained a model to see the change of the sigma, and added one curve figure to Appendix A.5. The sigma for OSCC is about -1.3e-5, for PNR is about -3e-5, for SCOD is about 0.01, while the loss value of SCOD is about 1e3 times over PNR and OSCC. Sigma is just differentiable but global.

---

> ### Author Response · Authors · 2023-11-23
> **Response to Reviewer hzuR (Part 2/2)**
>
> **W1.[This is absolutely not the same as having a scalable proxy for human evaluation. To make that claim, one would need to run a study comparing representations fine-tuned with human feedback to those fine-tuned with these hand-object interaction tasks.]**\
> We apologize for the confusion. We did not intend to make readers interpret the proposed method as a way of achieving alignment without human labels. The key idea we aim to convey is that the proposed approach takes inspiration from cognitive science, which posits that humans learn to extract a generalizable behavior representation from perceptual input by learning about a multitude of simpler perceptual skills that are critical for everyday scenarios and formalize this idea through the lens of multi-task learning where each “task” is a perceptual skill. Not only do each of these perceptual skills define a human prior about what is helpful for manipulation tasks, but they are each associated with large and well-labeled video datasets pioneered by the computer vision community.
>
> **W2. [It would be valuable to know why one would prefer a fine-tuned video model on these interaction tasks, as opposed to directly predicting them from the outset.]** \
> Good question, directly predicting affordance and visual representation are both major topics in robot learning. Predicting affordance can be more direct and intuitive for policy learning, which is highly correlated with action space. Visual representation learning like R3M, also has the advantage of flexibility, these representations can easily be used in any robot learning tasks with observation space, across the vision-language model, diffusion policy, and so on.
> We believe predicting from the outset is a good way for many scenes, but visual representation learning is also crucial, even some scenes like adroit pen task without accurate affordance, or other scenes with too many affordances in multi-task, direct predicting the outset may not cover all the affordance accurately, the hidden visual representation can also benefit.
>
> **W3. [Move the fanuc manipulation dataset to appendix]** \
> We will move it to the appendix in our final version.
>
>
> \
> Thanks again for your time and effort! For any other questions, please feel free to let us know.

---

> > ### Comment · Reviewer_hzuR · 2023-12-04
> >
> > Thanks to the authors for providing the extra experiments and clarifications!
> >
> > I still have some concerns about the framing of the paper around "human-oriented" when it is really about hand-object tasks being useful for fine-tuning. A couple of other specific replies:
> >
> > > [What about ablating each loss type separately? Please provide ablations for each individual task head]
> >
> > These results are pretty interesting, and suggest there is a lot of heterogeneity for how well the multiple tasks actually work for learning a useful visual representation. assembly is the only task that really benefits from training on all three tasks at once, and it is quite interesting that the fusion significantly hurts performance for relocate relative to *any* individual loss. Given that the discussion period is now over, the authors cannot provide a response for this, but I would like to see some interpretation in a follow up version of the paper.
> >
> > > Q3.[“Also, different vision tasks should have information interaction, for the human-like synesthesia.” This does not really add anything. What is human-like synesthesia?]
> > Human-like synesthesia is the way humans perceive spatial, temporal, and other information about the world simultaneously. In a task trajectory, humans gather information at the same time, like object localization, time to happen contact.
> >
> > This is not the meaning of synesthesia, so I would strongly recommend removing this from the paper.
> >
> > > W1.[This is absolutely not the same as having a scalable proxy for human evaluation. To make that claim, one would need to run a study comparing representations fine-tuned with human feedback to those fine-tuned with these hand-object interaction tasks.]
> > We apologize for the confusion. We did not intend to make readers interpret the proposed method as a way of achieving alignment without human labels. The key idea we aim to convey is that the proposed approach takes inspiration from cognitive science, which posits that humans learn to extract a generalizable behavior representation from perceptual input by learning about a multitude of simpler perceptual skills that are critical for everyday scenarios and formalize this idea through the lens of multi-task learning where each “task” is a perceptual skill. Not only do each of these perceptual skills define a human prior about what is helpful for manipulation tasks, but they are each associated with large and well-labeled video datasets pioneered by the computer vision community.
> >
> > What I am struggling with is comments like the following in the manuscript: "To mitigate this, another line
> > of works attempts to leverage human priors by explicitly involving a human in the learning loop to iteratively guide the representation towards human-orientated representations (Bobu et al., 2021; Katz
> > et al., 2021; Bobu et al., 2022; 2023a). However, these methods do not scale when learning from raw
> > pixels due to the laborious human costs. Our idea fills the gap between unsupervised/self-supervised
> > and human-guided representation learning." This implies that the presented method is a proxy for human evaluations at scale.
> >
> > Furthermore, it isn't clear that the specific tasks proposed in this paper are the most "human-aligned" of the possible fine-tuning tasks one could use for Ego4D. For example, localizing objects in the scene spatially (as opposed to hands, or the specific object in the hand) could also arguably be "human-aligned", or even predicting the future of how the objects will move (predictive coding being a very popular theory in cognitive science). The particular choice of tasks here is more about aligning representations to affordances (as in Bahl et al). I still think the paper would be much more compelling if a comparison to fine-tuning with other kinds of tasks was performed.
> >
> > The citation of Bahl et al was added to the updated manuscript. Thank you for adding it! However, I do not think the explanation of the pros / cons is well thought out in the single sentence: "(Bahl et al) adopts a multi-task structure for affordance. Compared with directly predicting affordance ,
> > the visual representation learning method is more flexible to fit various kinds of robot learning tasks
> > with observation space." I did not understand what this meant -- don't all robot learning tasks have an observation space? I think the explanation makes sense in the author reply, but a more concrete example of this is needed in the paper (or a particular experiment highlighting this in the paper would be even better!)
> >
> > For these reasons, and because I agree with other reviewers about the missing details for methodology for reproducibility, I am keeping my score.

---

### Official Review · Reviewer_uRTQ · 2023-10-31

**Soundness:** 2 fair
**Presentation:** 1 poor
**Contribution:** 3 good
**Rating:** 3
**Confidence:** 3

**Summary:**

A common visual representation for diverse manipulation tasks can be trained by fine-tuning a pre-trained visual representation on a small set of well-chosen, surrogate perceptual tasks that are related to or part of general manipulation tasks. Motivated by this intuition, this work proposes the Task Fusion Decoder (TFD), a transformer-style decoder that can be grafted on top of pre-trained vision encoders. The resulting system is trained, in a supervised fashion, (apparently) end-to-end on three different surrogate tasks involving classification, localization and detection of crucial state changes in egocentric videos of human manipulations (Ego4D).

**Strengths:**

This work addresses the important question of how to meaningfully learn general-purpose visual representations for manual interaction. A central idea is to leave behind conventional self-supervised training and leverage human examples of manipulation, and train in a supervised fashion on state transitions, which are arguably crucial aspects of any manipulation. The system leverages an existing, labeled dataset (Ego4D), but, thanks to training on surrogate tasks, generalizes to a broad class of manipulation tasks. The concept is not tied to any particular vision encoder; the paper showcases three different instances. The results indicate a clear performance increase in most cases compared to the original, pre-trained encoder.

**Weaknesses:**

The intuition of the paper and the grand lines are largely clear but I have spent considerable time trying to understand crucial details. The following statements describe my conclusions, referring to R3M, MVP, and EgoVLP as three possible "backbones" as the paper does:
- The encoder part of a pre-trained backbone is connected as input to the TFD.
- For training, the system is trained end-to-end on each of the three surrogate tasks simultaneously, training the weights of the TFD from scratch while fine-tuning the encoder in the process.
- For testing, the fine-tuned encoder is connected to its conventional downstream network (as applicable). The resulting network is then trained (keeping the encoder frozen?) and evaluated in its conventional fashion on manipulation benchmarks. The TFD has no role.

I am not entirely sure that my above understanding is correct. To arrive there, Appendix A.1 was essential reading. The number one improvement this paper should undergo is to clarify (in the main body) how exactly the backbone is connected to the TFD, how exactly the "fine-tuning" is done, and how the benchmark networks are set up, trained, and tested.

Another weakness is the evaluation. The TFD comprises a lot of computational machinery and $K$ stacked blocks (where $K$ is left undocumented) and involves many design choices, but the ablation study merely looks at the importance of the "time-related" and "spatial-related" tasks - and leaves the reader guessing which exactly these are, and what exactly the numbers in Table 3 mean. Even so, the results are not overly convincing, as one of the five environments performs worse under the TFD, and a second does not benefit. In contrast, it would be interesting to know the impact of the two attention mechanisms (as these implement the core motivation of this work) and how many of these blocks are required.

The paper tends to oversell a bit. As far as I can see, leveraging surrogate tasks in this manner is a novel idea but I'd hesitate to call it a "paradigm shift". It is said to "consistently" improve performance over the raw backbones, but the results are in fact mixed; on some problems performance actually decreases.

Figures 2, 3, 9 and 10 are crucial for understanding the paper but leave room for improvement. It is not obvious how the various inputs and outputs connect; the notation is not consistent everywhere. For example, is $j=10$ in Figure 2? Figures 2, 9 and 10 unnecessarily use different visual styles, making it difficult to see which differences are important and which are not. Figure 2 would probably be easier to understand if there was just one TFD block instead of the abstraction in the middle and the zoomed view on the right.

**Questions:**

- If it is true that the TFD has no role after training ("fine-tuning"), I'm not convinced that this is the best possible setup. The idea is to force the latent representation formed by the backbone encoder to generalize across tasks, but it is not clear how much of this generalization power lies within the considerable computational machinery of the TFD itself and is thus no longer available after fine-tuning. I'd like the authors to comment on this.
- The wording in Sec. 3.2 suggests there is exactly one state change frame. Why can't there be more than one?
- My two main concerns are lack of clarity and inconclusive results. At least the former can perhaps be fixed; I can perhaps be convinced to upgrade my ratings.

---

> ### Author Response · Authors · 2023-11-23
> **Response to Reviewer uRTQ (Part 1/2)**
>
> We would like to thank the reviewer for taking the time to review our paper and provide insightful suggestions.
>
> **Q1. [If it is true that the TFD has no role after training ("fine-tuning"), I'm not convinced that this is the best possible setup. The idea is to force the latent representation formed by the backbone encoder to generalize across tasks, but it is not clear how much of this generalization power lies within the considerable computational machinery of the TFD itself and is thus no longer available after fine-tuning. I'd like the authors to comment on this.]** \
> Good questions. We agree that TFD has no role after training, as it backward the gradients to the last few layers of the encoder to learn good representations (we froze most of the front layers of the encoder). Also, it is a common phenomenon to freeze the vision encoder in the visual-motor control community [A, B] for robotics policy learning, it can avoid finetuning a large vision model during policy training. Therefore, finding a strong enough visual representation is a core problem for this area.
> Our main contribution to TFD is to induce human perception style by multi-task structure for the existing vision backbones, and forming a new representation to test on policy learning. Thus, during finetuning, the last 1/3 layers of the original visual backbone are unfrozen, jointly training with TFD. The generalization is kept across tasks, as shown in the benchmark results in Fig.7. We also included a clear backbone setting in Appendix Section A.4.
>
> [A]Suraj Nair, Aravind Rajeswaran, Vikash Kumar, Chelsea Finn, and Abhinav Gupta. R3m: A universal visual representation for robot manipulation. arXiv preprint arXiv:2203.12601, 2022.
>
> [B]Tete Xiao, Ilija Radosavovic, Trevor Darrell, and Jitendra Malik. Masked visual pre-training for motor control. arXiv preprint arXiv:2203.06173, 2022.
>
> **Q2. [The wording in Sec. 3.2 suggests there is exactly one state change frame. Why can't there be more than one?]** \
> In this paper, we used the hand and object interaction of the dataset in the Ego4D data set, which only has one state change frame label. However, our pipeline is not constrained by the number of state change frames: if there is more than one state change frame, we can segment the whole video sequences with several parts or with more state change frames, and feed them into our network.
>
> **Q3. [My two main concerns are lack of clarity and inconclusive results. At least the former can perhaps be fixed; I can perhaps be convinced to upgrade my ratings.]** \
> We would like to thank the reviewer for the feedback! We have modified the method section 3.2 to make it clearer and added one additional figure in Appendix A.4 to further explain the training and testing setting of our paper. Please see our updated paper, thanks!
>
> **W1. [The number one improvement this paper should undergo is to clarify (in the main body) how exactly the backbone is connected to the TFD, how exactly the "fine-tuning" is done, and how the benchmark networks are set up, trained, and tested.]** \
> Thank the reviewer for the feedback! The backbone is connected to the TFD by spatial and temporal features, which are extracted from the visual encoder, and these features are used for cross-attention with different task embedding in TFD. For the “fine-tuning,” we unfreeze the last 1/3 layers for visual encoders, like gulp, we train the last four layers’ video transformer blocks of a total of 12 layers and train the TFD from scratch. For policy learning, we follow the traditional visual-motor control method, freezing the vision encoder we have learned and learning a policy learning network by a total of 20000 steps, every 1000 steps we evaluate the policy results.

---

> ### Author Response · Authors · 2023-11-23
> **Response to Reviewer uRTQ (Part 2/2)**
>
> **W2. [ The ablation study merely looks at the importance of the "time-related" and "spatial-related" tasks - and leaves the reader guessing which exactly these are, and what exactly the numbers in Table 3 mean. Even so, the results are not overly convincing, as one of the five environments performs worse under the TFD, and a second does not benefit.]** \
> We apologize for the confusion. The time-related tasks are the OSCC and PNR tasks, which contain temporal information, while the spatial-related task is the SCOD task, which contains spatial information. We also make a clearer ablation study by separating the “time-related” tasks into the oscc task and pnr task, here we provide ablation studies with three different tasks in TFD.
>
> |env                     | R3M+OSCC | R3M+PNR | R3M+SCOD | R3M+OSCC+PNR+SCOD|
> | :--------------------------------------------- | :-----: | :------------: | :---------: | :---------: |
> |micro | 21.00 |**28.00**|26.00|**28.00**|
> | light  | 82.00 | 69.00 | 70.00 | **83.00** |
> | ldoor| **39.00** |34.00 | 23.00 | 32.00 |
> | assembly|84.67|81.33| 83.33| **92.67** |
> | relocate| 39.33| **40.00**| **40.00**| 36.67 |
>
> The reason why some results are even worse in TFD is that in some environments maybe the three tasks we added are not the best combination, only one task can lead the policy learning in this simulation to the best result. Our TFD is a flexible multi-task structure, it can flexibly leverage different task combinations for the representation adjustment, but as seen in the table below, in most situations, adding all the tasks is the best combination.
>
>
> **W3. [The TFD comprises a lot of computational machinery and K stacked blocks (where K is left undocumented) and involves many design choices. In contrast, it would be interesting to know the impact of the two attention mechanisms (as these implement the core motivation of this work) and how many of these blocks are required.]** \
> As shown in the caption of Figure 2, the design for self-attention in each layer is for the interaction of each task, we hypothesize each task should have information change, for cross-attention, it is a connection between vision encoder backbone and TFD, which can decode vision representation from encoder into multi-task with human labels. In our original paper, the choice for the TFD is 2 layers, each layer has one self-attention and one cross-attention. And also, we add more experiments about different choices of K, as shown in the table below.
>
> | env              |      K=2           | K=3 | K=4 |
> | :--------------------------------------------- | :-----: | :--------: | :------------: |
> | micro |       **28.00**      | 23.00 |     25.00   |
> | light |       **83.00**       | 67.00 | 69.00 |
> | ldoor | 32.00 | **39.00** | 27.00 |
> | assembly | **92.67**|  88.67  |    94.00    |
> | relocate | 36.67 |   **44.67**  |   32.00     |
>
> After searching the parameter K, we find that the k doesn’t influence the results much, and K=2 with our original setting, maintains a good result in most environments.
>
>
> **W4. [it would be interesting to know the impact of the two attention mechanisms (as these implement the core motivation of this work)]** \
> As the caption for Fig. 2, the self-attention is designed mostly for multi-task tokens information exchange, we hypothesize that multi-task can benefit each other during training, and Table 4 has proven it by a better result on joint task finetuning with TFD. Cross-attention is a bridge from visual representation to TFD, which can decode different tasks’ information for the visual representation during finetuning for the last 1/3 layers of the visual encoder.
>
>
> **W5. [Unclear expression and figures]** \
> Thanks for pointing it out! We have deleted the strong words like “paradigm shift” and “consistently”. Fig. 2 is more important for understanding, the Fig. 9 and Fig. 10 are just specific TFD designs for different visual backbones, resnet, vit, and video transformers, which cover the main-trend computer vision backbone. We have a zoom-in for more clarity on the three general tasks. We also change the Fig. 2, making it more readable.
>
> We hope that our response has addressed your concerns, and turns your assessment to the positive side. If you have any questions, please feel free to let us know during the rebuttal window. We appreciate your suggestions and comments!

---

### Author Response · Authors · 2023-11-23
**General Response: Contributions and New Experiments**

We sincerely appreciate all reviewers’ time and efforts in reviewing our paper. We are glad to find that reviewers generally recognized our contributions:
* **Method.** Tackling an important problem for the community about how to adapt large pre-trained models to be useful for downstream tasks such as robotics [hzuR]; Leveraging human tasks hidden information for robotic manipulation with available human data [kBXX, UxCD, hzuR, uRTQ]; Model structure is flexible, and easily to be adapted by most research groups [hzuR, UxCD].
* **Experiments.** Offering a comprehensive set of cross-simulator experiments and real-world domain tests based on various models[kBXX, uRTQ, hzuR]; Indicating a clear performance increase [uRTQ, hzuR].
* **Writing.** Having the paper clearly presented and easy to follow [hzuR].


We also thank all reviewers for their insightful and constructive suggestions, which helped a lot in further improving our paper. In addition to the pointwise responses below, we summarize supporting experiments added in the rebuttal according to reviewers’ suggestions.

**New Experiments.**
* Ablative experiments on the layer numbers(K) of TFD [uRTQ];
    | env              |      K=2           | K=3 | K=4 |
    | :--------------------------------------------- | :-----: | :--------: | :------------: |
    | micro |       **28.00**      | 23.00 |     25.00   |
    | light |       **83.00**       | 67.00 | 69.00 |
    | ldoor | 32.00 | **39.00** | 27.00 |
    | assembly | **92.67**|  88.67  |    94.00    |
    | relocate | 36.67 |   **44.67**  |   32.00     |
* Decode tasks on robot policies and ego4d dataset in appendix A.6 [UxCD] ;
* Explore the change of sigma in the whole training process in appendix A.5 [hzuR];
* Ablative experiments on each individual task head [hzuR,uRTQ];
    |env                     | R3M+OSCC | R3M+PNR | R3M+SCOD | R3M+OSCC+PNR+SCOD|
    | :--------------------------------------------- | :-----: | :------------: | :---------: | :---------: |
    |micro | 21.00 |**28.00**|26.00|**28.00**|
    | light  | 82.00 | 69.00 | 70.00 | **83.00** |
    | ldoor| **39.00** |34.00 | 23.00 | 32.00 |
    | assembly|84.67|81.33| 83.33| **92.67** |
    | relocate| 39.33| **40.00**| **40.00**| 36.67 |
* Gather error bars for different camera views; [UxCD]

  | env              |     R3M          | R3M+ours|
  | :--------------------------------------------- |   :-----: | :--------: |
  | assembly | 93.67(3.77)|  **98.67**(1.89)  |
  | bin | 44.67(8.01)|  **56.33**(1.89)  |
  | hammer |**92.67**(5.66)|  86.34(3.30)  |
  | pen | 67.73(3.52)|  **70.00**(2.69)  |
  | relocate | 63.33(5.45)|  **66.22**(5.09)  |
* We release the dataset at [dataset_link](https://www.dropbox.com/scl/fi/tw7m6i7cgapefap7eozb3/fanuc_manipulation.zip?rlkey=53lz46vopb5r8r9l2d6d3jegt&dl=0), and the code at [code_link](https://www.dropbox.com/scl/fi/bdny032hk9q5jmed4doln/human_oriented_code.zip?rlkey=d8g3uvqr8t3s1kyz8edn7s8t1&dl=0).

We hope our pointwise responses below can clarify all reviewers’ confusion and alleviate all concerns. We thank all reviewers’ time again. Thanks!

---

### Meta-Review · Area_Chair_mppz · 2023-12-05

**Metareview:**

The paper proposes to use three human-object interaction related tasks to finetune pre-trained visual representations for robotics. Experimental results show that this multi-task finetuning can generally improve downstream performance in different simulator environments and also real-world robot tasks.

The paper received divergent scores from four reviewers. Reviewers uRTQ and UxCD had major concerns on the clarity of the paper and heuristic choices in modeling, and recommended rejection. Reviewer hzuR and kBXX leaned towards acceptance, but also mentioned that the paper lacks comparison with prior work using hand/object related tasks for robots. In addition, the performance improvements are not consistent across environments with the three tasks and there are no clear explanations. The rebuttal did not fully address the reviewers’ concerns.

Therefore, the AC decided to reject the paper.

**Justification For Why Not Higher Score:**

The paper lacks clarity and is hard to follow. The performance improvements are not consistent across downstream tasks and there are no clear explanations. It should also compare with prior work using hand/object related tasks.

**Justification For Why Not Lower Score:**

N/A

---

### Decision · Program_Chairs · 2024-01-16

Reject